



# Occurrence and growth of sub-50 nm aerosol particles in the Amazonian boundary layer

Marco A. Franco[1,2], Florian Ditas[2,a], Leslie A. Kremper[2], Luiz A. T. Machado[1,2], Meinrat O. Andreae[2,3], Alessandro Araújo[4], Henrique M. J. Barbosa[1], Joel F. de Brito[5], Samara Carbone[6], Bruna A. Holanda[2], Fernando G. Morais[1], Janaína P. Nascimento[7,b], Mira L. Pöhlker[2], Luciana V. Rizzo[8], Marta Sá[7], Jorge Saturno[2,c], David Walter[2,9,d], Stefan Wolff[2], Ulrich Pöschl[2], Paulo Artaxo[1], and Christopher Pöhlker[2]

[1]Institute of Physics, University of São Paulo, São Paulo 05508-900, Brazil
[2]Multiphase Chemistry Department, Max Planck Institute for Chemistry, 55128 Mainz, Germany
[3]Scripps Institution of Oceanography, University of California San Diego, La Jolla, CA 92037, USA
[4]Empresa Brasileira de Pesquisa Agropecuária (Embrapa) Amazonia Oriental, CEP 66095-100, Belém, Brazil
[5]IMT Lille Douai, Institut Mines-Télécom, Université de Lille, Centre for Energy and Environment, F-59000 Lille, France
[6]Federal University of Uberlândia, Uberlândia-MG, 38408-100, Brazil
[7]National Institute for Amazonian Research, Manaus, AM, 69.060-000, Brazil
[8]Federal University of Sao Paulo, Department of Environmental Sciences, Diadema, Brazil
[9]Department of Biogeochemical Systems, Max Planck Institute for Biogeochemistry, 07701 Jena, Germany
[a]now at: Hessian Agency for Nature Conservation, Environment and Geology, 65203 Wiesbaden, Germany
[b]now at: NOAA Global Systems Laboratory, Boulder, CO, 80305, US
[c]now at: Physikalisch-Technische Bundesanstalt, 38116 Braunschweig, Germany
[d]now at: Climate Geochemistry Department, Max Planck Institute for Chemistry, 55128 Mainz, Germany

**Correspondence:** Marco A. Franco (marco.franco@usp.br), Christopher Pöhlker (c.pohlker@mpic.de)

**Abstract.** New particle formation (NPF), referring to the nucleation of molecular clusters and their subsequent growth into the cloud condensation nuclei (CCN) size range, is a globally significant and climate-relevant source of atmospheric aerosols. Classical NPF exhibiting continuous growth from a few nanometers to the Aitken mode around 60-70 nm is widely observed in the planetary boundary layer (PBL) around the world, but not in central Amazonia. Here, classical NPF events are rarely

observed in the PBL, but instead, NPF begins in the upper troposphere (UT), followed by downdraft injection of sub-50 nm ($CN_{<50}$) particles into the PBL and their subsequent growth. Central aspects of our understanding of these processes in the Amazon have remained enigmatic, however. Based on more than six years of aerosol and meteorological data from the Amazon Tall Tower Observatory (ATTO, Feb 2014 to Sep 2020), we analyzed the diurnal and seasonal patterns as well as meteorological conditions during 254 of such Amazonian growth events on 217 event days, which show a sudden occurrence of particles

between 10 and 50 nm in the PBL, followed by their growth to CCN sizes. The occurrence of events was significantly higher during the wet season, with 88 % of all events from January to June, than during the dry season, with 12 % from July to December, probably due to differences in the condensation sink (CS), atmospheric aerosol load, and meteorological conditions. Across all events, a median growth rate (GR) of 5.2 nm h⁻¹ and a median CS of 0.0011 s⁻¹ were observed. The growth events were more frequent during the daytime (74 %) and showed higher GR (5.9 nm h⁻¹) compared to nighttime events (4.0 nm h⁻¹),

emphasizing the role of photochemistry and PBL evolution in particle growth. About 70 % of the events showed a negative





anomaly of the equivalent potential temperature ($\Delta\theta_{e}'$) – as a marker for downdrafts – and a low satellite brightness temperature ($Tir$) – as a marker for deep convective clouds – in good agreement with particle injection from the UT in the course of strong convective activity. About 30 % of the events, however, occurred in the absence of deep convection, partly under clear sky conditions, and with a positive $\Delta\theta_{e}'$ anomaly. Therefore, these events do not appear to be related to downdraft injection and

suggest the existence of other currently unknown sources of the sub-50 nm particles.

## 1   Introduction

New particle formation (NPF) refers to the nucleation of nanometer-sized molecular clusters from gaseous precursors and their subsequent condensational growth (e.g., Kulmala et al., 2004; Dal Maso, 2005; Kirkby et al., 2011; Kulmala et al., 2012; Kerminen et al., 2018). Under favorable atmospheric conditions, the newly formed particles grow through condensation of

semi- and low-volatile gases as well as coagulation into the cloud- and, thus, climate-relevant size range with diameters, $D$, larger than ∼80 nm (Kerminen et al., 2018; Pöhlker et al., 2018). NPF has been observed worldwide in the course of ground-based observations in different environments, such as rural and remote continental areas, urban environments, the Arctic and Antarctica, marine areas, and mountain sites (Kerminen et al., 2018, and references therein). Its wide and frequent occurrence makes NPF a major source of aerosol particle number concentrations and cloud condensation nuclei (CCN) worldwide (e.g.,

Merikanto et al., 2009; Spracklen et al., 2008; Nieminen et al., 2018; Yli-Juuti et al., 2020).

In the long list of locations where NPF has been detected in the planetary boundary layer (PBL) (Kerminen et al., 2018), the Amazon rain forest is a remarkable exception since the characteristic 'banana plots' – starting at few nanometers and growing up to the Aitken-accumulation mode (see definitions in Kulmala et al., 2012; Kerminen et al., 2018) – have rarely been observed (e.g., Andreae, 2013; Rizzo et al., 2018; Wimmer et al., 2018). In particular, Rizzo et al. (2018) discussed the

occurrence of sub-50 nm particle growth events ('Amazonian banana plots'), which resemble the behavior of 'classical NPF', but starting at much larger initial diameters (> 20 nm). These 'Amazonian banana plots' were observed in 3 % of the 749 days and were associated mainly with convective downdrafts. It is worth noting that the detection of growth events depends on the methodology used. However, it is clear that in Amazonia these characteristic events are less frequent.

This striking contrast to other environments has inspired researchers to investigate the underlying mechanisms that could

explain the absence of NPF as well as alternative particle sources that sustain the Amazonian aerosol population. Reasons for the absence of NPF in the PBL could be:

1. A suppression by isoprene (e.g., Kiendler-Scharr et al., 2009; Kanawade et al., 2011; McFiggans et al., 2019; Yli-Juuti et al., 2020), which is the most abundant volatile organic compound (VOC) in the Amazonian atmosphere (e.g., Andreae et al., 2018; Yáñez-Serrano et al., 2020).

2. The very low concentrations of inorganic precursor gases such as sulfur dioxide ($SO_2$, being converted into sulfuric acid $H_2SO_4$) as well as the bases ammonia ($NH_3$) and amines ($NR_3$) (Andreae et al., 1990; Trebs et al., 2004), which play key roles in the binary $H_2SO_4$–$H_2O$ and ternary $NH_3$–$H_2SO_4$–$H_2O$ nucleation mechanisms (Kirkby et al., 2011).





3. The high levels of relative humidity (RH), which have been associated with a low occurrence of NPF (e.g., Bonn and Moortgat, 2003; Hamed et al., 2011; Hyvönen et al., 2005).

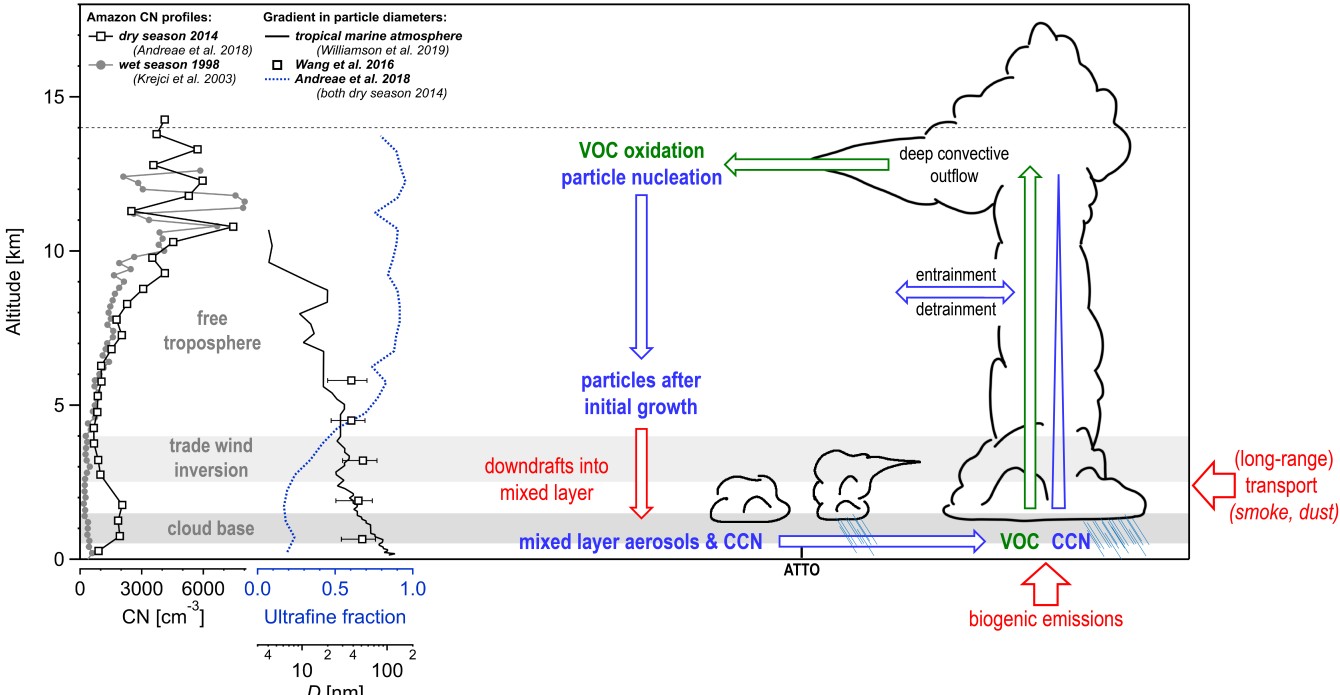

**Figure 1.** Conceptual scheme of sources, redistribution, and removal of aerosol particles and cloud condensation nuclei (CCN) over the Amazon. The scheme is inspired by previous conceptual illustrations by Krejci et al. (2003), Andreae et al. (2018), and Williamson et al. (2019). It further integrates experimental data by Wang et al. (2016), Krejci et al. (2003), Andreae et al. (2018), Baars et al. (2012); Williamson et al. (2019); Lauer et al. (2021). The study by Williamson et al. (2019) investigated the particle formation in the marine atmosphere but has been included here due to the mechanistic analogy. The three main categories of aerosol sources in the Amazon are shown in red. Emphasized are the aerosol cycling in the course of deep convective clouds with upward transport of volatile organic compounds (VOC) and aerosol particles, new particle formation in the free troposphere, downward transport with initial particle growth, and eventually downward injection of sub-50 nm particles in the clouds.

The occurrence of NPF is strongly dependent on the local conditions of individual sites, including meteorology, biogenic emissions, and air pollution levels. Nieminen et al. (2018) emphasized that the NPF occurrence and growth rates (GR) show a geographically inhomogeneous distribution, indicating that the underlying mechanisms are as manifold as complex. Typical atmospheric GR ranges from 1 to 12 nm h$^{-1}$ (Yli-Juuti et al., 2020). Further, different meteorological conditions have been associated with the occurrence of NPF and particle growth. Specifically, photochemical reactions under daytime conditions have been regarded as a driving force for both, nucleation and condensational growth (e.g., Nieminen et al., 2018; Kerminen et al., 2018; Hamed et al., 2011; Ma and Birmili, 2015). This is in line with a significantly higher occurrence of NPF under





clear sky conditions, as observed for instance in Hyytiälä, Finland (Dada et al., 2017). In addition, an association between the occurrence of NPF and convective clouds has been observed at different marine to continental sites (e.g., Perry and Hobbs, 1994; Clarke, 1992; Waddicor et al., 2012; De Reus et al., 2001; Wehner et al., 2015). Evidence of this phenomenon in the Amazon has also been reported by Andreae et al. (2018). While different potential explanations have emerged, the exact

mechanisms, precursors, and the spatial distribution in the context of clouds have remained unknown (Kerminen et al., 2018).

NPF has likely been altered as a result of industrialization, when anthropogenic emissions started to influence the atmospheric composition of trace species. Relative to remote sites, rural and urban locations tend to show higher NPF frequencies (typically $10 - 30\%$ event days) and higher particle GR ($4 - 12\,\mathrm{nm\,h^{-1}}$) (Kerminen et al., 2018; Nieminen et al., 2018). This relates to the fundamental question whether atmospheric concentration of certain, mainly anthropogenically derived, species

such as $\mathrm{H_2SO_4}$, have to exceed certain thresholds for NPF to occur. Recent evidence of pure biogenic ion-induced nucleation under controlled laboratory conditions offers mechanisms for pre-industrial pathways of NPF (Kirkby et al., 2016; Rose et al., 2018; Zhao et al., 2020). Accordingly, environments with low anthropogenic influence are of particular interest to investigate processes under conditions that approximate a pre-industrial state of the atmosphere. Amazonia is an ideal outdoor laboratory for such investigations under pristine conditions (Hamilton et al., 2014; Pöhlker et al., 2018). Of particular relevance is the wet

season with its episodic occurrence of pristine periods, which allows studying atmospheric processes - such as the occurrence of sub-50 nm particles ($\mathrm{CN_{<50}}$) - under conditions that approximate a pre-industrial state of the rain forest atmosphere (Andreae et al., 2015). At the Amazon Tall Tower Observatory (ATTO) – which is located in mostly untouched rain forest regions and has become a landmark site for atmospheric research (Andreae et al., 2015; Pöhlker et al., 2019) – March to May represent the cleanest months of the year with about 10 % of the time being considered as pristine periods (Pöhlker et al., 2018).

Figure 1 illustrates the main sources of aerosol particles and CCN in Amazonia, which can be broadly grouped into the following three categories:

1. A spectrum of biogenic emissions from the rain forest ecosystem, including the release of primary biological aerosol particles (i.e., pollen, spores, bacteria, fragments) (e.g., Pöhlker et al., 2012; Huffman et al., 2012; Löbs et al., 2020; Prass et al., 2021) as well as the emission of biogenic VOCs followed by their atmospheric oxidation and conversion into

secondary organic aerosols (SOA) (e.g., Chen et al., 2015; Liu et al., 2016; Saturno et al., 2018a; Leppla et al., 2021).

2. Long-range transport of transatlantically advected African dust and pollution (e.g., Pöhlker et al., 2018; Nascimento et al., 2021; Moran-Zuloaga et al., 2018; Holanda et al., 2020) as well as regional biomass burning smoke (Artaxo et al., 2013).

3. Driven by deep convective clouds, biogenic VOCs are transported into the upper troposphere ($\sim$10 km) where VOC

oxidation, nucleation of new aerosol particles, and initial particle growth occurs, which is fostered by low temperatures and a low preexisting aerosol surface area (Krejci et al., 2003; Andreae et al., 2018). Subsequently, the freshly formed particles are mixed downward into the PBL, where they continue to grow to CCN-relevant sizes (e.g., Krejci et al., 2003; Wang et al., 2016).





Several studies provide experimental and modeling support for the broad atmospheric relevance of UT particle production and the subsequent vertical mixing of the $CN_{<50}$ (e.g., Krejci, 2003; Krejci et al., 2005; Wang et al., 2016; Andreae et al., 2018; Williamson et al., 2019; Leino et al., 2019; Zhao et al., 2020; Rizzo et al., 2018; Toledo Machado et al., 2021). Figure 1 shows the presence of a $CN_{<50}$ pool in the Amazonian UT during the wet and dry seasons as well as an increase in $D$ with decreasing altitude due to condensational particle growth. The downward motion of the $CN_{<50}$ can be driven by strong convective downdrafts or weaker downward motions in stratiform cloud regions (Wang et al., 2016). Zhao et al. (2020) recently suggested that pure organic NPF based on biogenic VOCs dominates above 13 km, whereas ternary NPF involving organics and $H_2SO_4$ dominate between 8 and 13 km. In addition, an increase of sub-50 nm particles was observed, in particular in the early morning hours, suggesting a connection between these increased concentrations with vertical transport and deep convective clouds, as well as with lightning density. Major mechanistic questions regarding the vertical transport of the $CN_{<50}$ remain open (e.g., Toledo Machado et al., 2021), however.

This study aims to identify and characterize the occurrence of particle growth events in the size range from 10 to 50 nm in the PBL of Central Amazonia. Based on long-term observations, including complementary in-situ aerosol and meteorological measurements and satellite data, this work provides a robust 6-year data set to characterize Amazonian particle growth events ('Amazonian banana plots') and to give new insight into the occurrence and potential sources of sub-50 nm particles and related weather conditions. The knowledge obtained here about the sub-50 nm particle growth events addresses an important gap in our understanding of the Amazonian aerosol life cycle and will help to constrain the CCN sources and properties in this globally important ecosystem.

## 2 Measurements and data analysis

### 2.1 The Amazon Tall Tower Observatory (ATTO) site

The Amazon Tall Tower Observatory (ATTO) is located 150 km northeast of Manaus, Brazil, in a forest reserve. Detailed descriptions of the site, its location, instrumentation, and scientific missions can be found elsewhere (Andreae et al., 2015; Pöhlker et al., 2019). At the ATTO site, the first aerosol measurements were initiated in 2011 (e.g., Pöhlker et al., 2012; Saturno et al., 2018b). Since 2014, multiple continuous measurements of physical and chemical particle properties have been established and gradually extended (e.g., Pöhlker et al., 2016, 2018; Holanda et al., 2020; Saturno et al., 2018a; Schrod et al., 2020).

### 2.2 Terminology

According to Pöhlker et al. (2016) we define the Amazonian seasons as follows: the wet season is spanning from February to May, followed by the wet to dry transition period (WtoD) including June and July. The dry season is spanning from August to November, followed by the dry to wet transition period (DtoW) including December and January. For the Amazonian sub-micron particle population, which is characterized by a multi-modal size distribution, we use the widely established terms



Aitken mode ($50 - 100\,\mathrm{nm}$) and accumulation mode ($100 - 1000\,\mathrm{nm}$) (Pöhlker et al., 2016; Toledo Machado et al., 2021). In addition, we introduce the term sub-50 nm mode, defined as particles between 10 and 50 nm. We avoid using the term nucleation mode for this particle population as this term is typically defined as particles $< 25\,\mathrm{nm}$ and refers to aerosol population relatively soon after nucleation (Kulmala et al., 2012; Nieminen et al., 2018). The sub-50 nm particles analyzed here, however, have experienced initial aging and growth to diameters between 10 and 50 nm already. As a short version, we use $\mathrm{CN}_{<50}$ to refer to the particle fraction in the sub-50 nm mode. For the particle number concentrations of the individual modes, we use the symbols $N_{<50}$, $N_{\mathrm{Ait}}$, and $N_{\mathrm{acc}}$. $N_{\mathrm{CN}}$ is defined as the median total particle number concentration.

## 2.3 Aerosol measurements

This study focuses on particle number size distributions (PNSDs) obtained from a Scanning Mobility Particle Sizer (TSI Inc., Shoreview, USA; classifiers: first model 3080, later 3082; DMA: 3081; condensation particle counter: CPC 3772) sampling from the 60 m inlet at the 80 m high so-called triangular mast (02° 08.602'S, 59° 00.033'W; 130 m a.s.l.) at the ATTO site. The inlet height was chosen to be approximately 30 m above the average canopy height, which enables measurements close to the canopy without direct contact with the largest trees. The SMPS is located in an air-conditioned laboratory container at the foot of the mast. Sample air is transported through a 25 mm stainless steel tube (finetron tubes, Dockweiler AG, Neustadt-Glewe, Germany) and dried to relative humidity (RH) below 40 %. An automatic regenerating silica gel adsorption aerosol dryer, as described in Tuch et al. (2009) was installed upstream of the instruments in 2014 and was replaced by a custom-built and automated condensation aerosol dryer in March 2020. For more detailed information on the aerosol measurements setup, see Andreae et al. (2015).

The SMPS measurements cover a particle size range from 10 to 400 nm and yield a temporal resolution of 5 min. The PNSD data covers almost six years, from February 2014 to September 2020, covering 1 596 measurement days, and comprise 426 272 sample runs in total. The data coverage of ~67 % over the entire time frame (i.e., Feb 2014 to Sep 2020) can be considered as a robust data foundation and statistical basis for the observations and conclusion presented here.

The sizing accuracy of the SMPS was frequently checked with monodisperse polystyrene latex particles. Additionally, the data quality was continuously verified by complementary measurements with a Condensation Particle Counter (CPC, model 5412, Grimm Aerosol Technik, Ainring, Germany) measuring the total particle number concentration ($N_{\mathrm{CN}}$) $>4\,\mathrm{nm}$. All particle data were visually inspected for malfunction and contamination and further corrected for standard temperature and pressure (STP, 273.15 K, 1013.25 hPa) as well as inlet transmission efficiency according to Moran-Zuloaga et al. (2018). PNSD data was used for the analysis if $N_{\mathrm{CN}}$ from SMPS and CPC agreed within 15 %.

$\mathrm{CN}_{<50}$ are particularly prone to diffusion losses at surfaces (e.g., the tube surfaces of the inlet lines) (von der Weiden et al., 2009). Accordingly, the generally sparse occurrence of $\mathrm{CN}_{<50}$ in the Amazon frequently raises questions about whether these results are (systematically) biased by unaccounted diffusion losses (e.g., in the 60 m long inlets). The observations outlined below suggest the absence of large and unaccounted-for particle losses in the size range that is particularly relevant here (i.e., 10 to 100 nm) and further indicate that the observed PNSDs correctly reflect the actual atmospheric aerosol distribution:





- The inlet and particle transport is optimized for high particle transmission efficiency and short residence time of the sample air. According to the particle loss calculator provided by von der Weiden et al. (2009) and corresponding sensitivity tests, the 50 % transmission efficiency of the inlet at the lower end of the PNSD is reached at $D_{50\%} \approx 8$ nm. All PNSDs in this study have been corrected for diffusional, sedimentation, and inertial losses according to von der Weiden et al. (2009).

- Experiments with the SMPS running at the 60 m inlet line and a separate and mobile CPC running without inlet lines at the height of 60 m directly on the tower agreed well, which underlines that no significant fractions of $CN_{<50}$ were lost in the inlet lines.

- Finally, the PNSDs with the sparse particle occurrence < 20 nm reported here agree well with results in previous studies (e.g., Gunthe et al., 2009; Rizzo et al., 2018).

## 2.4 Multi-modal log-normal fitting of PNSDs

Each measured PNSD was fitted by a multi-modal log-normal distribution function, according to Heintzenberg (1994):

$$f(D_p, D_i, N_i, \sigma_i) = \sum_{i=1}^{n} \frac{N_i}{\sqrt{2\pi}\ln(\sigma_i)} \exp\left\{-\frac{[\ln(D_p) - \ln(D_i)]^2}{2\ln^2(\sigma_i)},\right\}, \tag{1}$$

where $D_p$ is the particle diameter, $n$ is the number of aerosol size modes to be fitted (with $n \leq 3$, see Sec. 2.2). Each mode is characterized by 3 main parameters: the mode number concentration, $N_i$, the mode geometric median diameter $D_i$, and the mode geometric standard deviation $\sigma_i$. A script was developed – similarly to the procedure in Hussein et al. (2005) – to provide an automatic user-free decision algorithm to obtain the size modes according to the following steps:

1. The diameter with the highest particle number concentration was used as the start parameter for the mean diameter of the dominant mode, $D_{dom}$. The least-squares fit of $D_{dom}$ was constrained within the range of $-30\%$ and $+20\%$ of the start parameter.

2. For $D_{dom} \geq 100$ nm – corresponding to the accumulation mode being dominant – the mode with the second highest concentration was searched in the size ranges of the sub-50 nm mode, $D_{<50} \in [9, 50)$, and Aitken mode, $D_{Ait} \in [50, 100)$. For $D_{dom} < 100$ nm – corresponding to either the sub-50 nm or Aitken mode being dominant – the mode with the second highest concentration was searched in the size range of the accumulation mode ($D_{Acc} \in [100, 300]$). For the second and (if present) the third mode, the same fitting routine as for the first (dominant) mode was applied.

3. The geometric standard ($\sigma_i$) deviation of all modes was constrained within the range of 1.1 to 1.55, which was optimized for the ATTO conditions.

4. Subsequently, a joint optimization of the previously obtained fit parameters ($D_i$, $\sigma_i$, and $N_i$) for the modes was conducted. The procedure is developed by fixing two of the modes and leaving the third free so that its parameters are again optimized by minimization using the least-squares method. The optimization order in this process was to optimize the




sub-50 nm mode, then the accumulation mode, and, finally, the Aitken mode. In this case, all the free diameters of the modes could vary between $0.5D_i$ and $1.5D_i$. As a measure of fitting quality, the algorithm compares the integrated particle number concentrations calculated from the measured size distribution and from the fitted curve and obtains the $R^2$ value for each measure. Examples of fits can be seen in Figure S1.

5.  Comparisons between the integrated particle number concentration of SMPS measurements ($N_{\mathrm{conc,\ SMPS}}$) and log-normal fitted size distributions ($N_{\mathrm{conc},\sum\mathrm{n\text{-}modes}}$) were made to assure the quality of the fits. We considered only fits in which the agreement of $N_{\mathrm{conc,SMPS}}$ and $N_{\mathrm{conc},\sum\mathrm{n\text{-}modes}}$ returned $R^2 > 0.8$, which means that about $97\%$ of the data are covered by the developed mode fitting. Within this data set, on average, fits with $R^2 = 0.97$ were obtained, which yielded a linear fit of $N_{\mathrm{conc,SMPS}}$ and $N_{\mathrm{conc},\sum\mathrm{n\text{-}modes}}$ with $R^2 = 0.997$ (Figure S2).

## 2.5 Identification of particle growth events

We analyzed the occurrence and properties of *particle growth events* ('Amazonian bananas plots') in the sub-50 nm, and Aitken mode size range. Characteristic examples of such growth events are shown and discussed in Sect. 3. The growth event identification is based on the following main steps:

1.  All data were smoothed to eliminate single exceptionally high or low values to avoid possible bias due to short intense particle peaks or dips. Accordingly, a two-dimensional smoothing algorithm (moving average with moving windows of 25 minutes for time and 20 nm size for particle diameter) was applied.

2.  All PNSD data were divided into 24 h subsets.

3.  Particle growth event days were then automatically flagged based on the guidelines in Kulmala et al. (2012). These guidelines were slightly modified by increasing the size threshold for the initial growth event identification from 20 nm to 40 nm to account for the characteristics of the Amazonian banana plots and PNSDs.

4.  Further following Kulmala et al. (2012), the total particle number concentration of particle diameters $> 40$ nm was then subtracted from the total particle number concentration of particle diameters $10 \leq D_{\mathrm{p}} \leq 40$ nm. Positive values in the PNSDs are marked as regions of interest for the occurrence of $\mathrm{CN}_{<50}$ that could result in particle growth events. Days fulfilling these criteria are flagged as *particle growth event days*.

5.  This method is sensitive to the integral particle number concentration in the Aitken and accumulation modes, and their seasonal variation, which might result in false positive or false negative event flagging. To account for that, the results from the automated identification routine were visually inspected and potentially misinterpreted events were excluded from the analysis. The inspection followed the procedure described in Dal Maso (2005), in which a particle growth event is characterized by i) the appearance of a distinct new mode of particles in the PNSD, ii) the particle size is inside the sub-50 nm mode, iii) the mode prevails for more than one hour, and iv) it shows signs of growth in time.





## 2.6 Growth rate and condensation sink

The growth rate (GR) and condensation sink (CS) – both important physical parameters in the characterization of growth events – were calculated following the procedures of Dal Maso (2005) and Kulmala et al. (2012). The GR is defined as the rate at which the mean geometric diameter $D_\mathrm{p}$ of the $\mathrm{CN}_{<50}$ population changes linearly with time:

$$GR = \frac{\mathrm{d}D_p}{\mathrm{d}t} = \frac{\Delta D_p}{\Delta t} = \frac{D_{p_2} - D_{p_1}}{t_2 - t_1} \ [nm \ h^{-1}], \tag{2}$$

where $D_{p1}$ is the geometric diameter of the sub-50 nm mode obtained by the multi-modal fit at the beginning of the growth event at time $t_1$ and $D_{p2}$ is the geometric diameter at the end of the growth event at time $t_2$. Thereby, the *beginning* of a growth event is defined as the moment at which $D_p$ starts to increase. The *end* of a growth event is reached when either (i) $D_p$ ($10 - 50$ nm) stops to grow, or (ii) the growth is interrupted due to sudden changes in air masses, or (iii) $D_\mathrm{p}$ reaches the Aitken mode – in this case, we selected $D_2$ as the last observed growth $D_p$ inside the sub-50 nm mode. There were a few events in which the growth stopped for a while and, afterward, restarted again. In these cases, we considered the second growth as a new growth event. The growth events considered in this study have a duration of at least one hour.

A moving average smoothing filter was applied at the mean geometric diameter interval $D_{p1} \leq D_p \leq D_{p2}$ and the fit was obtained by applying a linear model fit at the referred diameter interval. The model returned the following parameters: $R^2$, p-value, and GR. To assure the data quality during the analyses, we selected $R^2 > 0.6$ and p-value $< 0.05$.

The CS was calculated from the particle number concentration as (Dal Maso et al., 2002):

$$CS = 2\pi D \int_{D_{p,min}}^{D_{p,max}} D'_p \beta_m(D'_p) n(D'_p) \mathrm{d}D'_p = 2\pi D \sum_{D'_p} \beta_m(D'_{p,i}) D'_{p,i} N_i \ [s^{-1}] \tag{3}$$

where $N_\mathrm{i}$ is the particle concentration at the diameter $D'_{p,i}$ of the *i-esm* size bin; $D$ is the diffusion coefficient of the precursor condensable vapor, and $\beta_\mathrm{m}$ is the transition-regime correction (Fuchs and Sutugin (1971)), defined as:

$$\beta_m = \frac{1 + K_n}{1 + 1.677K_n + 1.333K_n^2} \tag{4}$$

which depends on the dimensionless Knudsen number, $Kn = 2\lambda/D_p$. The $Kn$ parameter represents the ratio of two length scales, where $\lambda$ is the effective mean free path of the vapor molecules in the gas (Dal Maso et al., 2002).

Physically, CS is a parameter that quantifies the ability of particles to remove condensable vapors from the atmosphere, incorporating them into the particle population and directly influencing the particle growth process. In this study, CS was calculated assuming $D = 0.117 \ cm^{-2} \ s^{-1}$, i.e., the value for sulfuric acid ($H_2SO_4$) (Gong et al., 2008), which is commonly used in the literature, allowing comparisons to other studies. We used the therm $CS_{growth}$ as the average CS during the growth particle event.

## 2.7 Meteorological parameters measurements

The meteorological parameters air temperature ($T$), incoming shortwave radiation ($SW$), rainfall ($P_{ATTO}$), air pressure ($p$), and relative humidity (RH) were measured at an 80 m high tower (02° 08.647'S, 59° 59.992'W; 130 m a.s.l.) located approximately





100 m from the ATTO aerosol mast. The measurements performed at the 80 m tower ranged from 2013 to 2018. Specifically, $SW$ and $P_{\text{ATTO}}$ were measured at the top of the tower, whereas $T$, $p$, and $RH$ were measured at 55 m, 55 m, and 81 m, respectively. From January 2019 to September 2020, the meteorological parameters air temperature ($T$), rainfall ($P_{\text{ATTO}}$), air pressure ($p$), and relative humidity (RH) were measured at the 321 m ATTO Tall Tower with a compact weather station

(Lufft, WS600-LMB, G. Lufft Mess- und Regeltechnik GmbH, Fellbach, Germany). Overall, meteorological parameters span the time frame from May 2013 to September 2020. Furthermore, an optical fog sensor (OFS, Eigenbrodt GmbH, Königsmoor, Germany) measured the near-field visibility since October 2015 at the height of 50 m. Fog occurrence is defined as visibility below 5000 m, which represents a threshold for light fog. Detailed information on the meteorological instruments can be found in Andreae et al. (2015).

## 2.8   Equivalent potential temperature

Variations of the equivalent potential temperature, $\theta_e$, have been used as a proxy to indicate downdraft occurrences (Machado et al., 2002; Betts et al., 2002; Rizzo et al., 2018; Wang et al., 2016; Gerken et al., 2016). $\theta_e$ quantifies the temperature of an air parcel, when lifted to a certain height where it condenses (characterized by its lift temperature, $T_L$), releasing the latent heat, and lowered adiabatically to 1000 hPa. In this study, $\theta_e$ was calculated with meteorological parameters measured in-situ and

was analyzed similarly to Wang et al. (2016) and Rizzo et al. (2018), using the definition described in Bolton (1980) as:

$$\theta_e = T_k \left(\frac{1000}{p}\right)^{0.2854(1-2.8\times10^{-4}r)} \exp\left[\left(\frac{3.376}{T_L} - 0.00254\right)r(1 + 8.1\times10^{-4}r)\right] [K],\tag{5}$$

$$T_L = \frac{1}{\frac{1}{T_k - 55} - \frac{\ln\left(\frac{RH}{100}\right)}{2840}} [K],\tag{6}$$

where $T_k$ is the ambient temperature in Kelvin, $p$ and $r$ are the ambient pressure ( hPa), and the water mixing ratio ($g/kg$),

respectively, and $T_L$ is the lifting condensation level temperature in Kelvin. To obtain the variations in $\theta_e$, it was necessary to subtract seasonality and diurnal variations.

The steps of this process are illustrated in Figure S3 and are described as follows: $\theta_e^{'}$ was obtained by subtracting the mean seasonal trend values, in which we considered both wet and dry seasons, for each year of the time series. Then, the calculated mean diurnal cycle of $\theta_e^{'}$ was subtracted from $\theta_e^{'}$, at the same time of the day, resulting in a new time series: $\Delta\theta_e^{'}$. The quantity

$\Delta\theta_e^{'}$ is the anomaly in $\theta_e$, and represents the deviation of $\theta_e$ from its expected value for that time of the day and season. Values of $\Delta\theta_e^{'} < 0$ are a proxy to the occurrence of downdrafts and indicate a decrease in $\theta_e$ due to air masses from the free troposphere that enter the PBL, typically related to the occurrence of rain Wang et al. (2016); Rizzo et al. (2018). Other processes may also be related to a decrease in $\theta_e$, such as evaporation of rainfall, river breeze, and advection mechanisms.



## 2.9 GOES-16 cloud brightness temperature

This study uses infra-red brightness temperature ($Tir$) data obtained by the Geostationary Operational Environmental Satellite (GOES), GOES-16, from November 2017 to April 2020. This data set comprises a total of 914 days with measurements every 10 min, as an indication of the troposphere's meteorological conditions. The Advance Baseline Imager (ABI) – a state-of-the-art 16-band radiometer on board GOES-16 – was employed in this study, specifically, Band 13, the infrared window at $10.3\mu m$. These measurements are less sensitive than other infrared bands to gas absorption, which allows estimating the cloud-top brightness temperature. An area of 3x3 pixels centered at ATTO was selected for obtaining the time series of $Tir$, representing around $6.0 \times 6.0\,\text{km}^2$. Meteorological conditions related to shallow clouds/clear sky are described by a warm $Tir$ and deep convection conditions by a cold $Tir$. Here, we considered $Tir > 280$K as a nearly clear sky condition, $245 \leq Tir < 280$K corresponding to shallow clouds and cumuliform clouds, and $Tir < 245$K to all convective clouds associated with deep convection (Machado and Rossow, 1993; Machado et al., 2002). $Tir$ in nearly clear sky conditions corresponds roughly to the temperature in the PBL.

## 3 Results and discussion

### 3.1 Particle number size distributions for wet and dry season

In agreement with previous studies, our long-term PNSD measurements showed the distinct characteristics of the Amazonian wet and dry season aerosol populations (e.g., Roberts et al., 2001; Gunthe et al., 2009; Artaxo et al., 2013; Pöhlker et al., 2016; Rizzo et al., 2018). We chose a different representation of the typical PNSD shapes in Figure 2 by showing them as frequency distributions (FDs). The PNSDs differ significantly between both seasons: during the wet season, clear Aitken and accumulation modes stand out, separated by a distinct Hoppel minimum (Hoppel et al., 1986). The Aitken mode is centered at 71 nm, the Hoppel minimum is centered at 102 nm and the accumulation mode is centered at 153 nm. In contrast, Figure 2b shows the typical dry season PNSDs characterized by a strong mono-modal shape with a dominating accumulation mode, reflecting the prevalence of biomass burning pollution (e.g., Rissler et al., 2006; Brito et al., 2014). The accumulation mode is centered at 146 nm. In addition, the overall distribution shape in Figure 2b reveals a contribution of the Aitken mode, centered at 68 nm and visible as a small shoulder on the dominant accumulation mode.

In terms of the abundance of $CN_{<50}$, the FDs in Figure 2 reveal that aerosols in this size range are rather sparse - though not absent - during both seasons. The occurrence of $CN_{<50}$ in central Amazonia along with the absence of 'classical' NPF, as it is detected, e.g., in the Scandinavian boreal forests and shown in e.g., Kulmala et al. (2004); Heintzenberg et al. (2017); Kerminen et al. (2018); Dall'Osto et al. (2018), is well documented in the literature (e.g., Roberts et al., 2001; Pöhlker et al., 2016; Rizzo et al., 2018). Although occurring sparsely, the episodic presence of $CN_{<50}$ causes a distinct mode below about 50 nm, noticeable in the wet season FDs in Figure 2a. However, the $CN_{<50}$ do not clearly show up in the corresponding mean and median PNSDs. It is further worth noting that the PNSDs under the *remote* rain forest conditions at ATTO as shown here differ significantly from PNSDs that were obtained in the rain forest atmosphere with an influence of the urban emission plume





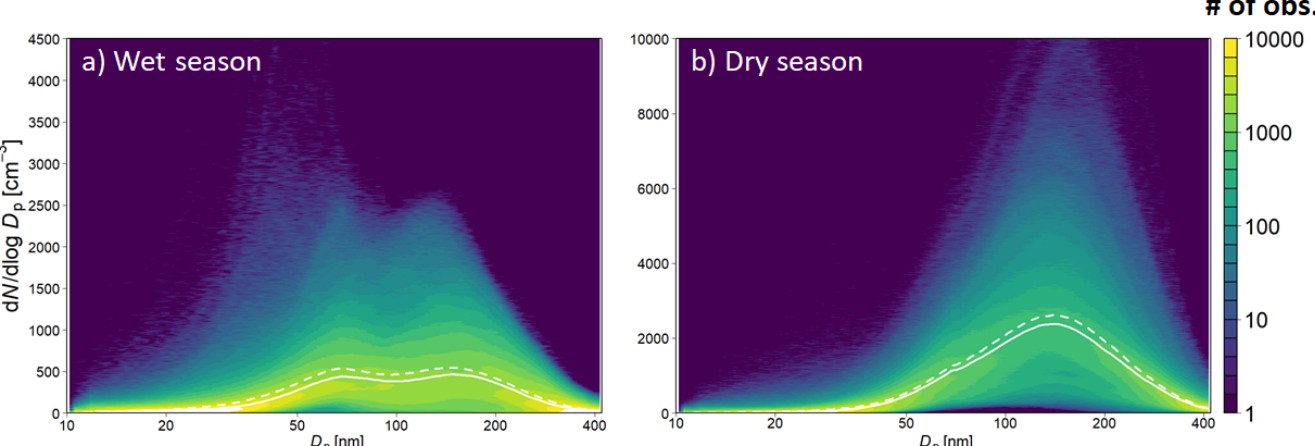

**Figure 2.** Frequency distribution (FDs) of particle number size distributions (PNSDs) for the Amazonian wet (a) and dry season (b). Data basis comprises 6.5 years of SMPS data from Feb 2014 to Sep 2020. Color code denotes the number of observations. Solid lines represent the median and dashed lines the mean PNSDs. Please note the different y-axes. The wet season PNSDs show pronounced Aitken and accumulation modes. An overwhelming accumulation mode dominates the dry season PNSDs. The wet and dry season FDs of PNSDs emphasize a comparatively sparse occurrence of $CN_{<50}$, which form a weak, though characteristic, mode below 50 nm.

from Manaus (e.g., Cirino et al., 2018; Fan et al., 2018; Wimmer et al., 2018; Glicker et al., 2019). These urban-influenced PNSDs are characterized by strongly enhanced particle concentrations below about 20 nm. As the mean and median PNSDs do not sufficiently reflect the abundance and properties of $CN_{<50}$ in Figure 2, the following paragraphs summarize the in-depth analysis that allowed us to extract their event characteristics, seasonal and diurnal variability, and their estimated significance.

## 3.2 Particle growth events characterization

The abundance of $CN_{<50}$, which show up as a weak, though noticeable, mode during the wet season (see the overall outline in Figure 2a), results from the episodic occurrence of $CN_{<50}$ events and their subsequent growth. Figure 3 shows a typical example of an 'Amazonian banana plot' representing two growth events, as often observed at ATTO. The first example in Figure 3 starts in the morning hours around 08:30 local time (LT) with an average initial diameter slightly larger than 30 nm. The particles grow for about four hours reaching the Aitken mode size range up to ∼ 60 nm. On the same day, a secondary growth event starts around noon with an average initial growth diameter slightly larger than 20 nm, growing during the afternoon hours. The initial diameters at the onset of the growth events in Figure 3 are well above the lower size limit of the SMPS (i.e., 10 nm), which implies that the event characterization is not distorted or limited by the effectively measured size range.

The growth events shown here resemble the events reported by Wang et al. (2016). Note that for all previous studies in Amazonia, the growth events were observed during the wet season, indicating that this event type is a typical wet season phenomenon associated with precipitation (Zhou, 2002; Wimmer et al., 2018; Rizzo et al., 2018). The procedure described in





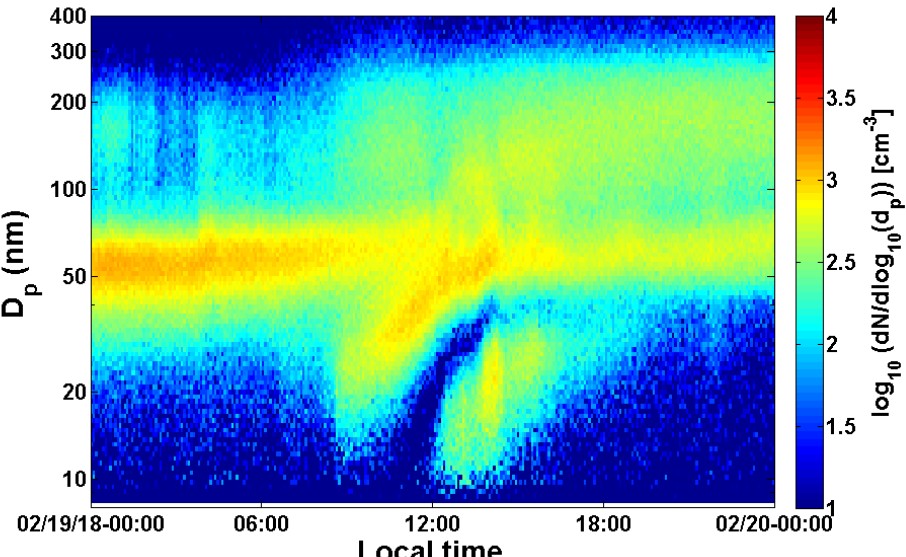

**Figure 3.** A characteristic example of a $CN_{<50}$ growth event at ATTO on 19 February 2018. The temporal evolution of the particle number size distribution (PNSD) is shown as a heat map, emphasizing the pronounced Aitken mode as well as a particle growth events from the sub-50 nm particle to the Aitken mode during daylight.

Sec. 2.5 returned 254 characteristic particle growth events on 217 days of the 1596 measured days, corresponding to a frequency of occurrence of ∼14 % of event days. For the entire measurement period (Apr 2014 – Sep 2020) this further corresponds to about 30 event days per year. The events have a clear seasonality, with more cases in the wet season, corresponding to ∼ 88 % of events from January to June, while in the dry season from July to December, only ∼12 % were observed. Additional aspects of seasonality are discussed in Sec. 3.3.

The GR frequency distribution in Figure 4a shows a clear peak centered around the median of 5.2 nm h$^{-1}$. The median GR of this study agrees well with the median GR of 5.5 nm h$^{-1}$ obtained by Rizzo et al. (2018). Figure S4a contrasts median GR obtained at different sites in Amazonia and worldwide and shows that the median GR of this study is within the GR ranges obtained at remote boreal (0.5 to 5.3 nm h$^{-1}$) or polar sites (0.2 to 5.5 nm h$^{-1}$). The $CS_{growth}$ frequency distribution in Figure 4b shows a clear peak centered around the median of 0.0011 s$^{-1}$. In contrast, the median CS calculated for the entire observation days is 0.0032 s$^{-1}$, which corroborates that particle growth events at remote sites are expected when CS values are low (Figure 6d). The Amazonian CS also agrees with what is observed in other remote regions (Figure S4 b). For example, boreal sites have average CS ranging from 0.00098 to 0.0039 s$^{-1}$ (Kerminen et al., 2018).

Figure 5a shows the FD of PNSDs exclusively for the periods of growth events, starting 3 hours before the event's onset and lasting until the time when growth stopped being observed (according to Sec. 2.6). Figure 5b shows the separated $CN_{<50}$, Aitken and accumulation modes for the median SD of Figure 5a. The characteristic multi-modal shape of the wet season PNSDs stands out. The FD for the growth events further underlines the sparse particle abundance below 20 nm. Figure 5b shows that





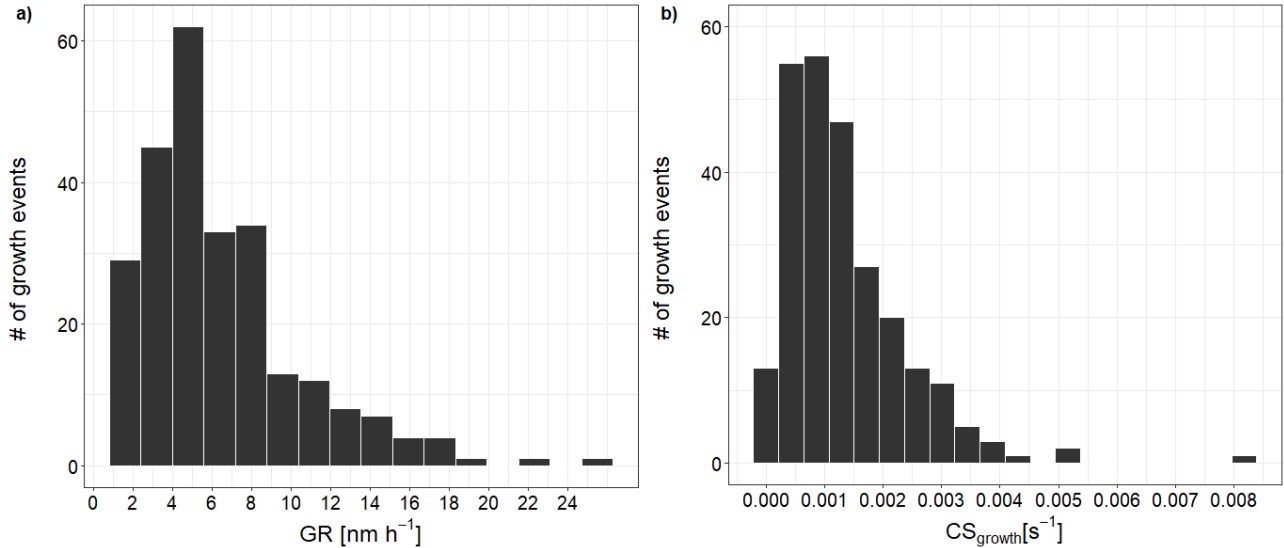

**Figure 4.** Histograms showing the frequency distribution of the growth rate, GR, **(a)** and condensation sink during the growth events, $CS_{growth}$, **(b)** for all observed 254 particle growth events.

the Aitken and accumulation modes are the dominant modes of the median PNSD. The $CN_{<50}$ mode, although small and not readily perceptible from the median PNSD, has a significant contribution, however, and is centered at $D_{CN_{<50}} = 37\ nm$ (mean), with $\sigma_{CN_{<50}} = 1.5$. For comparison, the diameters selected as the initial values for the growth events, $D_{p,i}$, have a median value of $26.1\ nm$, with 25th and 75th percentiles of $19.0$ and $33\ nm$, respectively. The Aitken mode is centered at $D_{AIT} = 67\ nm$ (mean), with $\sigma_{AIT} = 1.4$, and the accumulation mode is centered at $D_{ACC} = 170\ nm$ (mean), with $\sigma_{ACC} = 1.4$. This clearly shows that the initial diameter of growth events in the Amazon is typically larger than reported in other regions (Nieminen et al. (2018)).

### 3.3 Seasonality

The pronounced atmospheric seasonality in central Amazonia has been characterized by means of meteorological, aerosol, and cloud microphysical data in previous studies (e.g., Pöhlker et al., 2018, 2019; Moran-Zuloaga et al., 2018; Saturno et al., 2018a). Figure 6a shows the typical seasonality of $P_{ATTO}$. The highest rain rate occurs during the wet season, with $P_{ATTO}$ peaking in March and April, while the minimum in $P_{ATTO}$ occurs between July to September. A similarly pronounced seasonality can be found in various aerosol properties. The $N_{CN}$ in the size range between 10 to 400 nm had its minimum in the wet season months March and April with a median of $\sim 280\ cm^{-3}$ and its maximum in the dry season months August to November with a median of $\sim 1400\ cm^{-3}$ (Fig. 6c).

The same pattern can be found in the monthly median CS in Figure 6d. The strong seasonal differences in the physical aerosol properties - here manifested in a wide range of $N_{CN}$ and different PNSD shapes - have a substantial influence on the





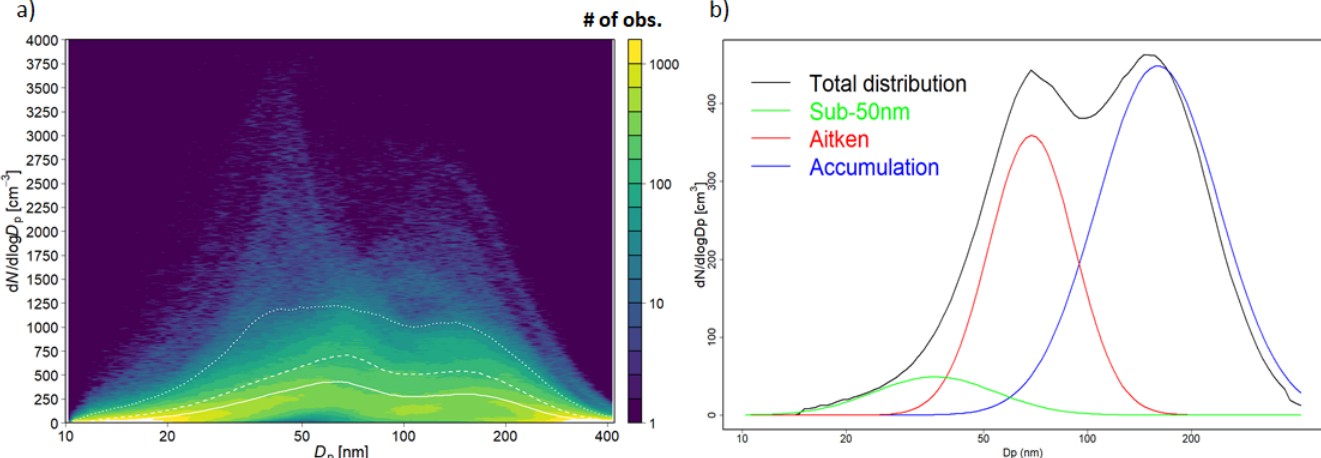

**Figure 5.** a) Frequency distribution of particle number size distributions exclusively for growth event periods. Color code denotes the number of observations, solid lines indicate the median and dashed lines the 75th and 90th percentiles. The PNSDs for the growth events show a high frequency of $CN_{<50}$ and Aitken size modes. The FDs do not show any evidence of *open* PNSDs towards the lower detection limit. b) Median PNSD (black line) of the 254 observed particle growth events, showing its 3 log-normal modes: in green, the sub-50 nm size particle mode; in red, the Aitken mode; in blue, the accumulation mode; and in black, the total distribution. The adjusted $R^2$ obtained for the calculated multi-modal log-normal fit is 0.99.

concentration of available cloud condensation nuclei (CCN) and, thus, cloud micro-physical processes in the Amazon Basin (Pöhlker et al., 2016, 2018; Lauer et al., 2021). These results agree well with long-term measurements at another central Amazonian site (i.e., ZF2 site) presented by Rizzo et al. (2018) and allow to put earlier campaign-wise measurements into a broader data context (e.g., Roberts et al., 2001; Roberts, 2003; Zhou, 2002; Rissler et al., 2004, 2006; Martin et al., 2010).

Figure 6e shows the seasonal pattern in the $CN_{<50}$ growth event frequency. We found the highest frequencies during the wet season, peaking in April with about 26 %, and dropping down during the transition period (WtoD, Jun and Jul) to a minimum with almost zero events in August. The growth event occurrence stayed remarkably low during the dry season months with frequencies mostly below 3 % from July to November. Frequency levels increase again during the transition period (DtoW, Dec and Jan). The seasonality in growth event occurrence corresponds well with the seasonality in monthly rainfall and appears inversely related to the seasonality in $N_{CN}$ and $CS$. This agrees with previous studies in the Amazon, suggesting a close link between generally low particle concentrations and the appearance of $CN_{<50}$ and Aitken mode particles in the PBL (e.g., Krejci, 2003; Wang et al., 2016). The low CS might further favor the characteristic growth patterns of these events through the condensation of semi- and low-volatile gaseous compounds on the $CN_{<50}$ particle fraction as available surfaces (see example in Fig. 3a) (Dal Maso, 2005; Dal Maso et al., 2007; Dada et al., 2017; Kerminen et al., 2018; Nieminen et al., 2018; Wiedensohler et al., 2019).

The $N_{CN}$, $CS$, and growth event occurrence in Figure 6 are all based on the same multi-year SMPS data set. The underlying data availability is documented in Figure 6b as the number of valid measurement days. Table 1 shows the GR and the median





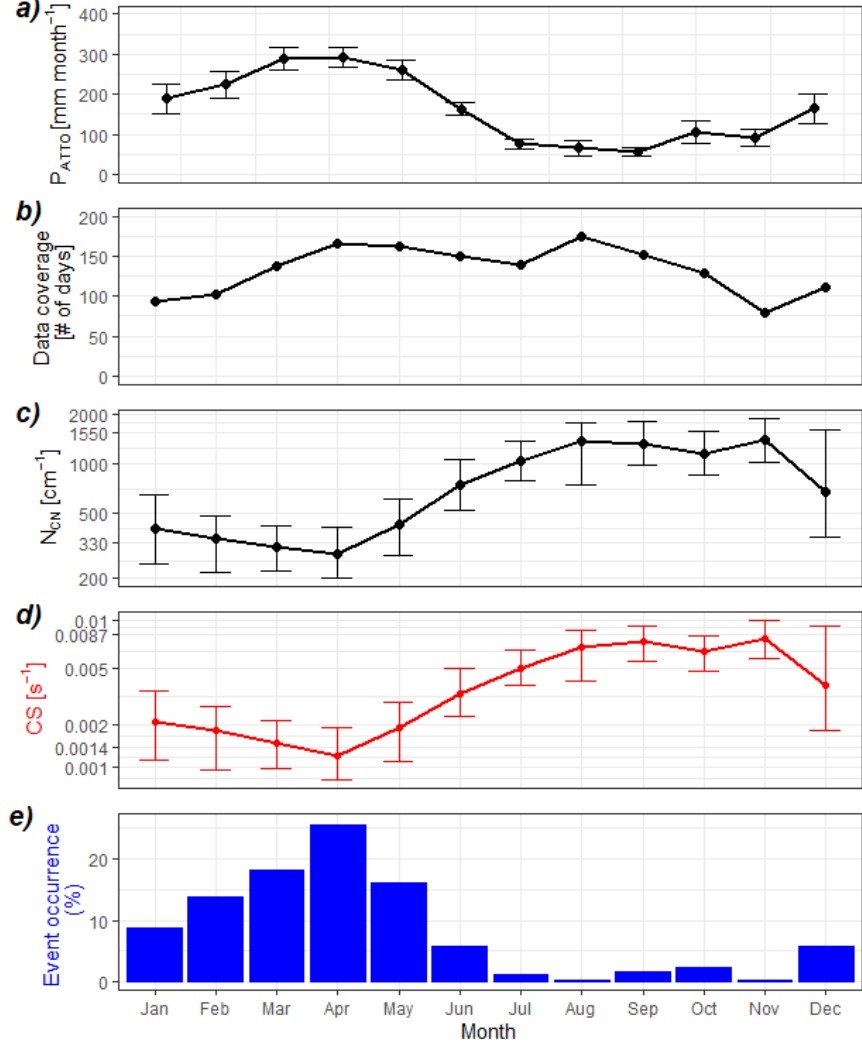

**Figure 6.** The seasonal cycles of selected meteorological and aerosol parameters presented as monthly averages for the entire observation period from February 2014 to June 2020. From top to bottom: a) $P_{\text{ATTO}}$ monthly rain measured at ATTO, where error bars denote the standard deviation. b) SMPS data coverage in the number of measurement days. c) Median total particle number concentration in size range of $10\,\text{nm} < d < 400\,\text{nm}$, calculated from SMPS data, error bars denote the interquartile range (please note the logarithmic scale). d) Median condensation sink for the month calculated from SMPS data, error bars refer to the interquartile range (please note the logarithmic scale). e) Annual cycle of the occurrence of $CN_{<50}$ particle growth events.

$CS_{\text{growth}}$ of the growth event for all the months. In the wet season, when ~88 % of all growth events occurred, the values of GR fluctuate around the median GR of $5.2\,\text{nm}\,\text{h}^{-1}$. The $CS_{\text{growth}}$ shows a similar behavior, fluctuating around $0.0011\,\text{s}^{-1}$. The lowest median GR of the wet season is found in March, with $3.7\,\text{nm}\,\text{h}^{-1}$, while the lowest $CS_{\text{growth}}$ of the wet season is found in April, with $0.0008\,\text{s}^{-1}$. The number of particle growth occurrences drops in the dry season, comprising only ~12 % of the total





growth events. If December is not included in the dry season, the percentage of dry season growth events decreases to $\sim 6\,\%$. The $CS_{growth}$ in the dry season indicates that the events occur on days with a cleaner atmosphere compared to the monthly median CS, as they are similar to the CS of the events observed in the wet season. The only exceptions are the events from October, whose median $CS_{growth}$ was $0.0044\,s^{-1}$.

**Table 1.** Monthly median, 25th, and 75th percentiles (in brackets, respectively) of GR and $CS_{growth}$. The percentage of growth events in each month is also presented.

| Month | GR [nm h$^{-1}$] | CS$_{growth}$ [ s$^{-1}$] | Fraction of events (%) | # of events |
|:---:|:---:|:---:|:---:|:---:|
| Jan | 5.0 (2.5 - 6.9) | 0.0012 (0.0008 - 0.0016) | 8.7 | 22 |
| Feb | 5.2 (3.9 - 8.1) | 0.0013 (0.0008 - 0.0021) | 13.8 | 35 |
| Mar | 3.7 (2.6 - 6.3) | 0.0010 (0.0005 - 0.0015) | 18.1 | 46 |
| Apr | 5.3 (4.5 - 8.6) | 0.0008 (0.0005 - 0.0012) | 25.6 | 65 |
| May | 5.5 (4.0 - 8.2) | 0.0011 (0.0006 - 0.0019) | 16.1 | 41 |
| Jun | 6.1 (4.6 - 9.4) | 0.0025 (0.0012 - 0.0030) | 5.9 | 15 |
| Jul | 8.2 (6.2 - 8.6) | 0.0034 (0.0026 - 0.0038) | 1.2 | 3 |
| Aug | 5.2 (5.2 - 5.2) | 0.0011 ( 0.0011 - 0.0011 ) | 0.4 | 1 |
| Sep | 10.4 (3.7 - 18.8) | 0.0017 (0.0012 - 0.0019) | 1.6 | 4 |
| Oct | 8.8 (4.9 - 13.1) | 0.0044 (0.0033 - 0.0052) | 2.4 | 6 |
| Nov | 4.2 (4.2 - 4.2) | 0.0037 (0.0037 - 0.0037) | 0.4 | 1 |
| Dec | 6.2 (4.7 - 8.0) | 0.0012 (0.0009 - 0.0015) | 5.9 | 15 |

The results show that particle growth events are likely related to CS, in which the higher the median monthly CS the lower the occurrence of growth events. Higher amount of particles aging by coagulation returns higher values of CS, which means a decrease in the lifetime of the precursor vapor in the atmosphere. In the dry season, regional and long-range transport biomass burning plumes are the main source of aerosols in the lower troposphere, which suppresses the smaller ones from the sub-50 nm mode to grow. With the increase in precipitation (thus, in the wet deposition), the number of particles of the
accumulation mode in the atmosphere decreases significantly. This influences on the decrease of CS, so that the occurrence of growth events increases again, as observed in December (DtoW transition).

### 3.4 Diurnal trends

The diurnal patterns of the growth event occurrence are shown in Fig. 7 in relation to meteorological parameters, such as air temperature, $SW$, $P_{ATTO}$, $RH$, and near-field visibility representing fog. Note that we contrasted the diurnal cycles for the
entire observation period (i.e., Feb 2014 to Sep 2020, shown as solid lines) with the wet season months (i.e., Feb to May, shown as dashed lines). In the diurnal cycles considering the whole period of observations, $T$ and $SW$ show the typical tropical rain forest conditions with about 12 h daylight and a remarkably low amplitude in $T$ spanning on average only $5°$ C. Rainfall is most intense in the afternoon hours, with the highest precipitation intensity at 15:00 LT, resulting in average annual rainfall of





around 2000 mm. The data also shows a secondary maximum in the early morning, which has been associated with nocturnal long-lived mesoscale systems (Toledo Machado et al., 2021). The $RH$ levels reach on average 100 % during the early morning and decrease during the day to around 75 %. Fog typically occurs in the second half of the night and often in the early morning before sunrise (i.e., between 03:00 and 07:00 LT), when $T$ is lowest. Sporadically, fog also occurs shortly after rain showers,

which is not reflected in the average conditions.

Figure 7e shows the diurnal cycle (blue line) of the median $CN_{<50}$ number concentration, $N_{<50}$, exclusively during the particle growth event days. For comparison, the median diurnal cycle of $N_{<50}$ comprising all measured PNSD is also presented (black line). The particle concentration on growth event days is significantly higher than that including all analyzed PNSDs, with median daily values of 64 (38 - 108) $cm^{-3}$, compared to 49 (29 - 81) $cm^{-3}$ for all days. The $N_{<50}$ diurnal cycle for growth

event days shows a strong increase from midnight to 09:00 LT, peaking at $N_{<50} = 88 \ cm^{-3}$, compared to $N_{<50} = 56 \ cm^{-3}$ at the same time for the total data. This suggests that $CN_{<50}$ are injected into the PBL by rainfall events during the late afternoon and early night and last until mid-morning. Similar behavior for $CN_{<50}$ has been reported by Toledo Machado et al. (2021).

The diurnal cycle of the growth event onsets has a rather broad maximum spanning over the early morning hours, from 06:00 to 10:00 LT. It peaks at about 07:00 LT and then gradually decreases towards noon (see Fig. 7e), which is in agreement with

what is observed in Figure 7f. A second local and less pronounced maximum is visible from 13:00 to 15:00 LT. The growth events reported during daytime likely correspond to rainfall events as reported by Toledo Machado et al. (2021) and probably the vertical transport of $CN_{<50}$ and Aitken size particles due to strong downdrafts in the course of convective rainfall, and the injection of these particle populations into the PBL as reported in Wang et al. (2016) and Andreae et al. (2018).

$P_{ATTO}$ shows two maxima: A pronounced and rather defined maximum in the early morning at around 07:00 LT, which

follows a gradual increase in precipitation during the second half of the night, and a broader maximum during the afternoon hours between 13:00 and 17:00 LT. Although about 74 % of events occur during the day, there are still ∼26 % that take place during night conditions, between 19:00 - 05:00 LT. In particular, the occurrence of growth events from 01:00 - 05:00, which represents about 16 % of the total observed events, is evidence for complexity in the causes and mechanisms of particle injection and growth.

The evolution of the PBL also has a strong influence on the diurnal pattern. At night, the nocturnal BL close to the forest canopy is decoupled from the residual layer above (Fisch et al., 2004). In the morning hours - as soon as convection becomes effective - air masses transported into and within the residual layer are mixed into lower levels and measured at the canopy level. Consequently, $CN_{<50}$ and Aitken mode particles advected with the residual layer will be mixed downwards and appear at the 60 m inlet in the morning hours, typically around 8:00 LT. Section 3.5 further discusses the meteorological conditions

regarding convective downdrafts and the atmospheric conditions under which the growth events are observed. Toledo Machado et al. (2021) discuss this daily mechanism of particle growth in more detail.

A contrast in GR and CS is observed when day and night events are compared, as shown in Figure 8. Daytime events, which correspond to ∼74 % of the events, have significantly higher GR and $CS_{growth}$, at 5.9 nm h$^{-1}$ and 0.0012 s$^{-1}$, respectively. The nighttime events, which account for ∼26 %, have GR and $CS_{growth}$ of 4.0 nm h$^{-1}$ and 0.0009 s$^{-1}$, respectively. To verify

the statistical significance of the difference between day and nighttime values, the Wilcoxon rank-sum test was applied. The



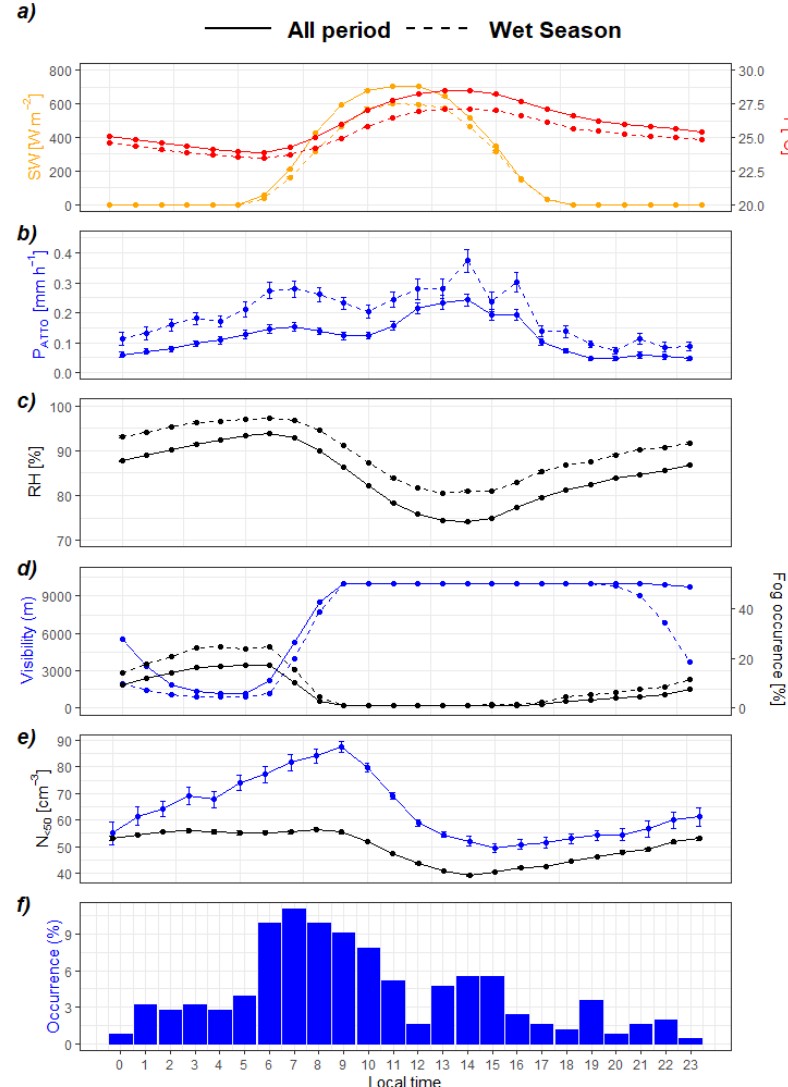

**Figure 7.** Diurnal cycle of selected meteorological parameters and the diurnal occurrence of particle growth events. From top to bottom: a) incoming shortwave radiation ($SW$, orange) and air temperature ($T$, red) at 26 m (canopy level) height. b) average local rain rate ($P_{\mathrm{ATTO}}$, blue). c) relative humidity (RH) at 26 m. d) visibility (blue) measured by a fog monitor and fog occurrence (black) with lines representing the first quartile, e) median diurnal cycle of $N_{<50}$ for all the data (black) and only for the days in which particle growth events were observed (blue). And f) the diurnal cycle of the particle growth event onsets. All error bars denote the standard error. The data shown represent all observations (full lines) and wet season subsets (dashed lines).

p-value obtained for GR is $3.6 \times 10^{-6}$, while the p-value obtained for $CS_{\mathrm{growth}}$ is 0.02, indicating that the data groups regarding day and night are statistically different considering a significance level of 0.05. The observed differences are likely due to the different atmospheric mechanisms during daytime and nighttime. Daytime events are directly influenced by sunlight, which





controls photochemistry and hence the oxidation of SOA precursors. Sunlight also drives the dynamics of the PBL. Other phenomena also play an important role in daytime events, such as the peaks of precipitation that coincide with the peaks of growth events. In contrast, nocturnal events may have different causes and mechanisms, which may be related to meteorology, entrainment of air and particles from the free atmosphere into the PBL, and perhaps the contribution of biogenic sources from

5    the surface.

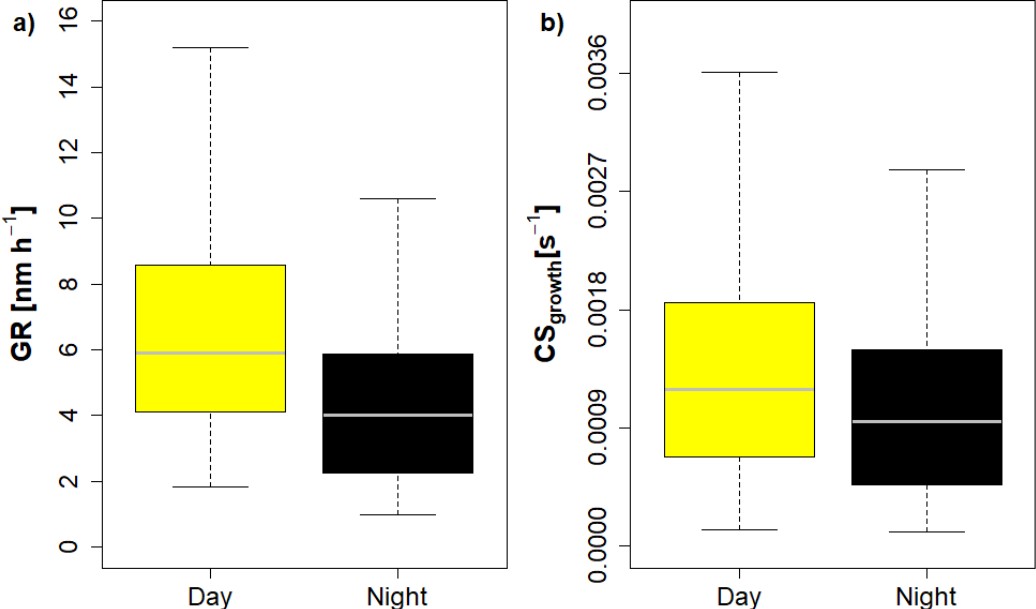

**Figure 8.** Boxplot of a) GR and b) $CS_{growth}$ related to growth events that occurred during the day or night. Nighttime events occured between 19:00 and 05:00 LT, while daytime events occurred between 06:00 and 18:00 LT. The box represents the quartiles, whiskers represent 90th and 10th percentiles, and the horizontal lines represent the median.

### 3.5    Meteorological and cloud conditions during growth events

This section explores the weather and convective transport conditions linked to particle growth events. Figure 9a shows a histogram of the growth event frequency as a function of the associated anomaly of the equivalent potential temperature ($\Delta\theta'_e$) at the onset of the growth events (see sec. 2.8). A negative $\Delta\theta'_e$ is an indicator for air mass downdrafts from higher altitudes,

10    as has been shown for the events analyzed by Wang et al. (2016). We found that ~63 % of events were likely associated with air mass downdrafts ($\Delta\theta'_e < 0$), whereas ~37 % were likely associated with $\Delta\theta'_e > 0$. We do not consider $\Delta\theta'_e = 0$ as it is a boundary region between meteorological conditions. Figure 9b shows a histogram of all growth events as a function of the associated cloud brightness temperature, which is an indicator for deep convective clouds ($Tir < 245\,K$) vs clear sky/shallow cloud conditions ($Tir > 280\,K$). For all events after 2017 (when $Tir$ data is available), we found that ~36 % were likely





associated with deep convective clouds ($Tir < 245\,K$, red), $\sim26\,\%$ with clear sky/shallow clouds ($Tir > 280\,K$, blue), and $\sim38\,\%$ with mixed sky conditions ($245\,K < Tir < 280\,K$, gray).

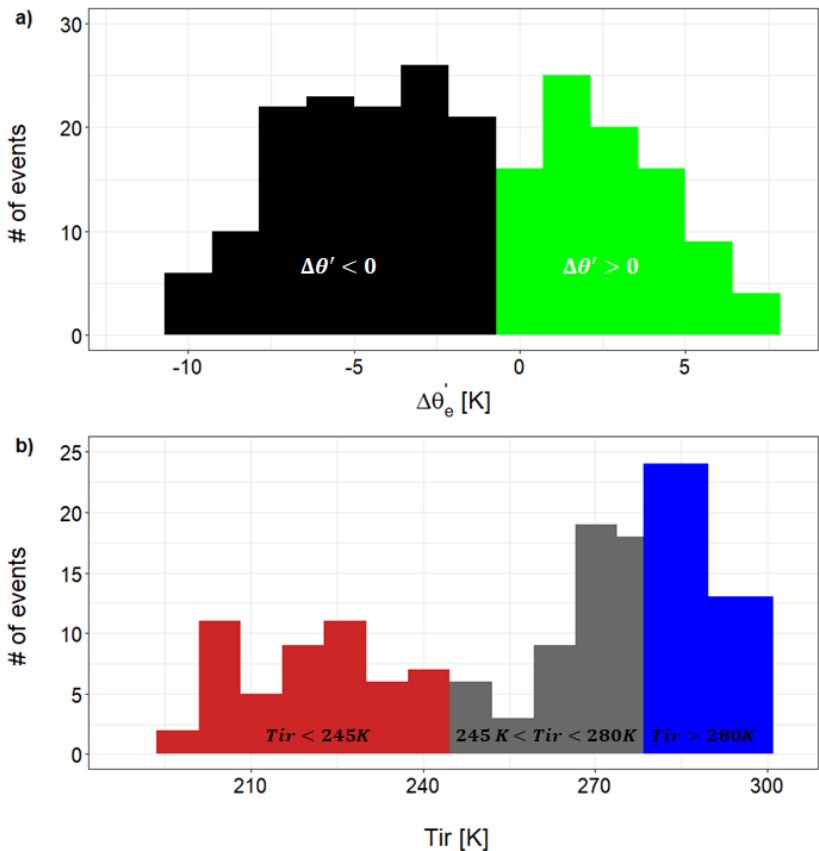

**Figure 9.** Histogram of the a) values of $\Delta\theta'_e$ at the beginning of the particle growth events. Black bars represent events during downdraft conditions and the green bars during undefined conditions, and the b) values of $Tir$ at the beginning of the particle growth events. Red bars represent events during deep convection conditions, blue during clear sky/shallow clouds conditions and gray during mixed sky conditions.

The most contrasting $Tir$ conditions were observed between the lower quartile ($Tir < 228\,K$), representing deep convective cloud conditions, and the upper quartile ($Tir > 281\,K$), representing clear sky conditions, with 36 events contained in each

5    group. As an example, four of these 'extreme' events were selected (Figure 10) in daytime and nighttime conditions. The growth events under clear sky conditions are characterized by a trimodal aerosol population, with accumulation and Aitken modes as well as a third mode below 50 nm. As an example, the daytime event on 14 March 2019 was characterized by the occurrence of a significant amount of $CN_{<50}$ during nighttime (starting after 00:00 LT), with a pronounced peak in concentration around 06:00 LT and with an onset of particle growth also at 06:00 LT, which lasted for about 12 h. These events could be associated

10    with advection processes, e.g., by a downdraft in the gust front (clear sky nighttime) or by nighttime rainfall and subsequent growth in the early morning by PBL processes.



The events under deep convective conditions – both, during daytime and nighttime – resemble the events reported by Wang et al. (2016). Here, downdrafted air masses in the course of deep convection inject $CN_{<50}$ particles from the upper troposphere into the PBL, followed by particle growth into the Aitken mode. In both these cases, the atmosphere is very clean, with low concentrations in the accumulation mode. Before the growth event, most of the particle population is in the Aitken size mode,

5 which is removed by the injection of upper or mid-tropospheric air during the downdraft event, so that only the $CN_{<50}$ aerosol population remained. Note that about 4 hours after the start of the growth event, the accumulation mode particles (re)appeared, probably by mixing with surrounding airmasses.

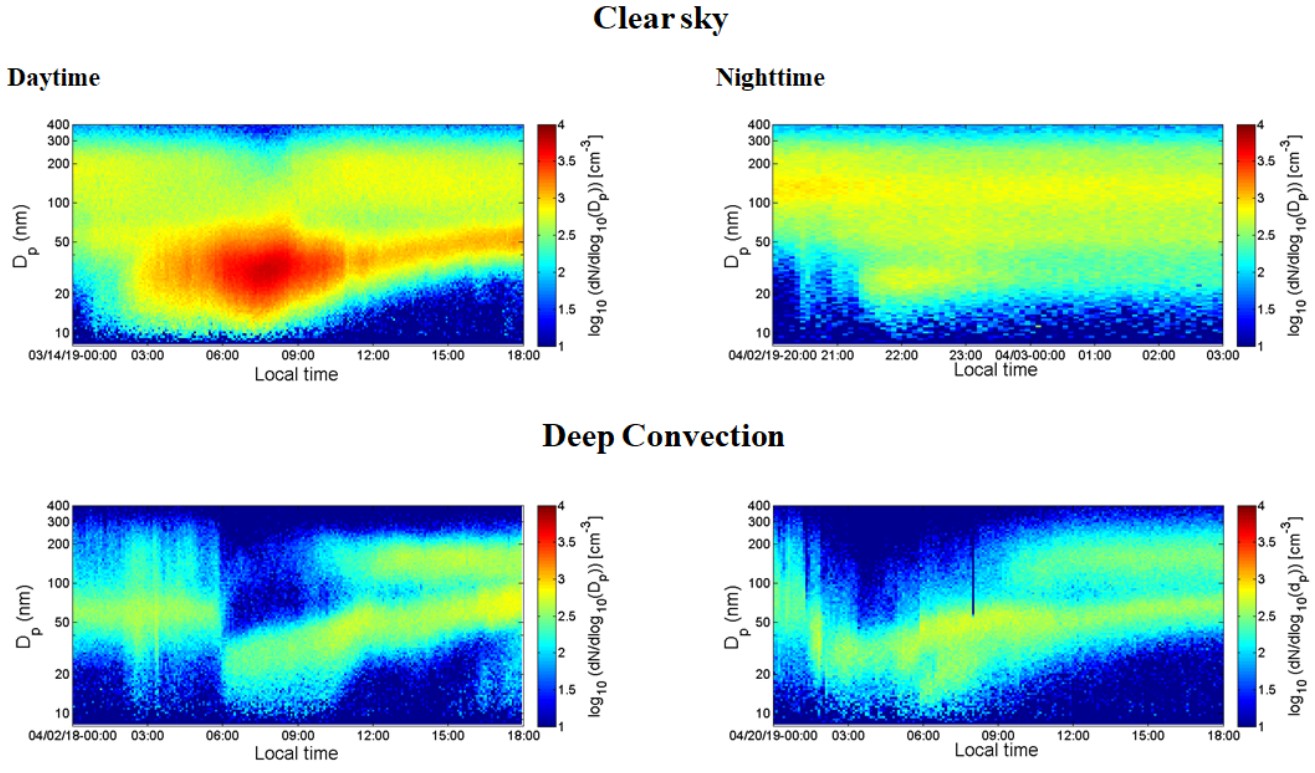

**Figure 10.** Selected particle growth events according to the $Tir$ value at the onset of the event. Events under clear sky/shallow cloud conditions during daytime (left) and nighttime (right) were selected based on $Tir > 281$ K (third quartile), while deep convection events were chosen based on $Tir < 228$ K (first quartile).

Figure 11 shows boxplots for GR in (a) and $CS_{growth}$ in (b) for clear sky and deep convection conditions. The median GR for clear sky conditions is $7.0 \, \mathrm{nm \, h^{-1}}$, whereas the median GR for deep convection conditions is $3.8 \, \mathrm{nm \, h^{-1}}$. Regarding CS,

10 under clear sky conditions the median is $0.0016 \, \mathrm{s^{-1}}$, while under deep convection, the median CS is $0.0005 \, \mathrm{s^{-1}}$. The results show that different meteorological processes play an important role for the different particle growth events observed. The events that occurred under deep convection conditions present lower $CS_{growth}$ and GR. Two main factors may influence this




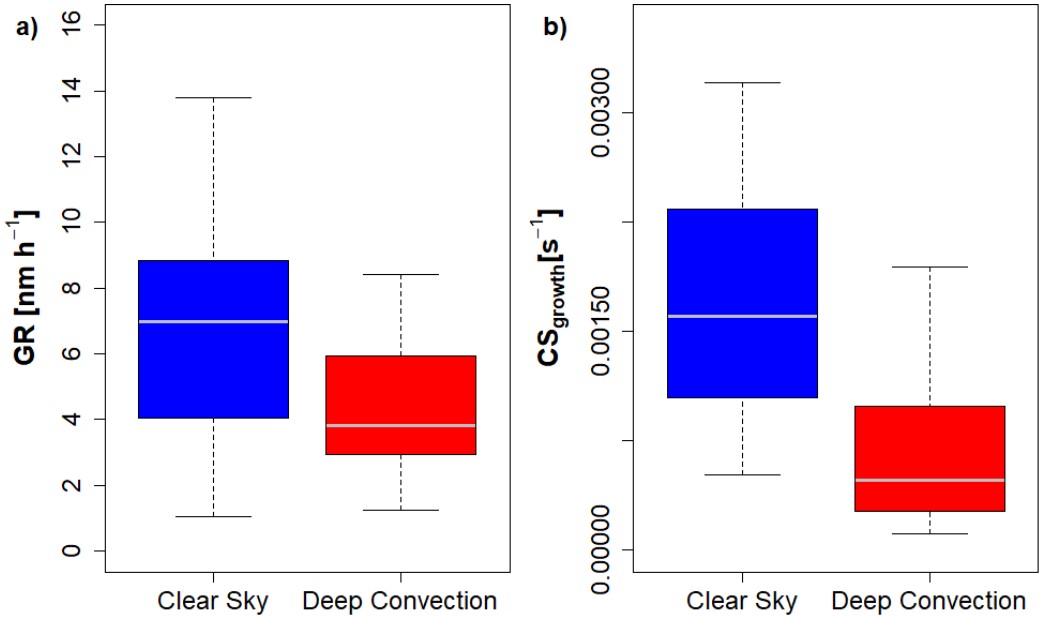

**Figure 11.** Boxplot of a) GR and b) $CS_{growth}$ for clear sky conditions ($Tir > 281\,K$) vs deep convection conditions ($Tir < 228\,K$). Boxes represent the quartiles, whiskers represent 90th and 10th percentiles, and the horizontal line represents the median.

result: the precipitation during deep convection conditions cleans the atmosphere by wet scavenging, resulting in lower $CS_{growth}$ values, and the presence of clouds, reduces the availability of sunlight and thereby suppresses photochemical production of condensable species.

The daily frequency distribution of growth events (see Figure 7f) was divided into four groups (G1 to G4), where G1 and
G4 represent the nighttime and G2 and G3 represent the daytime events. The daytime events were divided considering the occurrence of two frequency peaks: the first peak, representing 53 % of the growth events, is included in G2, covering the time from 06:00 to 11:59 LT. The second peak, representing 21 % of the growth events, is included in G3, covering the time from 12:00 to 17:59 LT. The nocturnal events were divided according to the possible state of the PBL. Group G4, with 10 % of the events, is defined from 18:00 to 00:59 LT, when the growth event population may still have some influence from late in the day,
since some convective events occur later in the day and early night, and the convective BL is collapsing at this time. Group G1, representing 16 % of the total growth events, covers the time from 01:00 to 05:59 LT, when the nocturnal BL is well established and different mechanisms such as air mass entrainment into the BL, nighttime rainfall events, or biogenic processes could play a role in the aerosol particle dynamics. Table 2 shows the median $Tir$, $\Delta\theta_e'$, the initial diameter at the onset of the growth event, $D_i$, and the GR and $CS_{growth}$ for the four hourly groups.

The results show a clear difference between the four groups (G1 to G4) regarding the $Tir$ and $\Delta\theta_e'$ conditions during the event onset. The nocturnal groups G1 and G4 have median $\Delta\theta_e'$ of -0.8 and 0, respectively. Their median $Tir$ indicates conditions closer to low clouds and clear skies, with median values equal to 269 K and 274 K for G1 and G4, respectively.





**Table 2.** Median, 25th and 75th percentiles (in parenthesis) for $Tir$, $\Delta\theta_e^{'}$, $D_i$, GR and $CS_{growth}$ for each hourly group. $Tir$, $\Delta\theta_e^{'}$, $D_i$ correspond to the onset of the particle growth event.

| Group | Hour (local time) | $Tir$ [K] | $\Delta\theta_e^{'}$ | GR [nm h$^{-1}$] | $CS_{growth}$ [s$^{-1}$] | $D_i$ [nm] | Fraction of events (%) |
|---|---|---|---|---|---|---|---|
| G1 | 01:00 - 05:59 | 269 (238, 283) | -0.8 (-4.6, 1.5) | 3.8 (2.3, 5.5) | 0.0009 (0.0004, 0.0015) | 27.7 (21.7, 35.3) | 16 |
| G2 | 06:00 - 11:59 | 268 (236, 282) | -1.9 (-5.3, 2.1) | 6.3 (4.2, 8.6) | 0.0012 (0.0007, 0.0020) | 27.7 (20.6, 33.5) | 53 |
| G3 | 12:00 - 17:59 | 245 (219, 283) | -3.6 (-6.8, 0.5) | 5.6 (4.0, 8.6) | 0.0012 (0.0008, 0.0015) | 24.6 (19.4, 29.9) | 21 |
| G4 | 18:00 - 24:00 | 274 (227, 276) | 0 (-4.3, 2.1) | 4.2 (2.1, 6.3) | 0.0010 (0.0006, 0.0015) | 27.8 (19.5, 32.7) | 10 |

The $Tir$ diurnal cycle (Figure 12) shows clear differences between the groups. Within the G1 period, $Tir$ shows a minimum at 04:00 LT, when the average brightness temperature reaches 256 K, indicating convective activity and early precipitation compared to all days.

The G4 group, with the smallest number of growth events, is not significantly different from the median diurnal cycle for
all days, suggesting that this 10 % of growth cases appears not be related to specific meteorological events. There are signs of convection at 16 LT, which coincides with the precipitation peak in the afternoon. Afterward, $Tir$ increases, going to clear sky conditions during the night. The median GR for G1 and G4 varies from 3.8 to 4.2 nm h$^{-1}$, respectively, and $CS_{growth}$ approximately constant around 0.0010 s$^{-1}$, while $D_i$ remains stable. The daytime groups G2 and G3 have the lowest median $\Delta\theta_e^{'}$ at the onset of the events, with -1.9 and -3.6, respectively, indicating that convective downdraft activity plays an essential
role during these growth events. The diurnal cycle of $Tir$ presents lower values during the whole day compared to the overall average, and the group G2 also shows minima in the early morning coinciding with the precipitation peaks, and also in the afternoon (Figure 7b).

The G3 group is the one with the most convective characteristics. The $Tir$ values indicate a strong occurrence of convective systems throughout the day, mainly in the afternoon. In particular, the pronounced decrease at 15 LT reaches deep convection
conditions, with $Tir$ = 249 K, approximately 1 hour earlier than expected considering the whole observation days. The median GR ranges from 6.3 to 5.6 nm h$^{-1}$ in G2 and G3, respectively, and the median $CS_{growth}$ remains constant, at 0.0012 s$^{-1}$. The smaller value of GR in G3 compared to G2 (diurnal events) could be associated with deep convection systems, which reduce the solar irradiance and thereby influence the photochemical processes. The presence of clouds has been associated with lowering GR and even lower occurrences of NPF and particle growth events (Dada et al., 2017; Kerminen et al., 2018). The
median $D_i$ for G3 is the lowest of the four groups, at 24.3 nm, indicating that the strong convective downdrafts are more effective in transporting smaller particles from the free troposphere into the PBL, which agrees with what has been observed previously (Wang et al., 2016). Therefore, the growth events of groups G2 and G3 are probably influenced by the strong convective systems during daytime. In particular, the G3 group has the most significant characteristics of deep convection and intense occurrences of downdraft throughout the day.

When discriminating the growth events by positive or negative $\Delta\theta_e^{'}$ at the event's onset (here defined as $\Delta t = 0$) and looking 10 hours before and after this time, the different behavior of $\Delta\theta_e^{'}$ and $Tir$ near the growth event is evident. Figure 13 shows the behavior of the mean ensembles of $\Delta\theta_e^{'}$ and $Tir$ around $\Delta t = 0$ for $\Delta\theta_e^{'}$ less than the 25th percentile (-5.3) and higher than





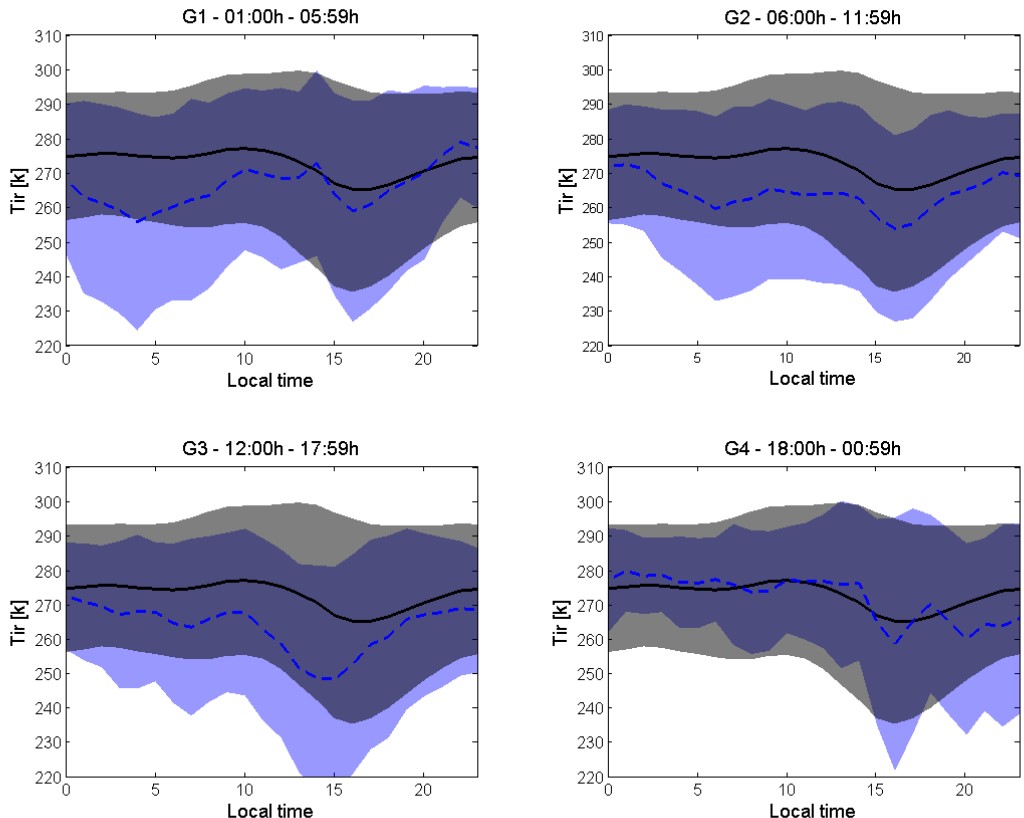

**Figure 12.** Average diurnal cycle of $Tir$ for the event days in the groups G1 to G4. The blue dashed line represents the average diurnal cycle of $Tir$ relating to the days when the particle growth events for a given group were observed. The black line represents the average diurnal cycle for all days on which PNSD measurements were made. The colored shadows represent the standard deviation.

the 75th percentile (+1.5) at the event's onset. Considering the case where $\Delta\theta'_e < -5.3$ at the event's onset, both $\Delta\theta'_e$ and $Tir$ strongly decrease from 10 h before the event and reach a minimum at $\Delta t = 0$, where $\Delta\theta'_e = -7.6$ and $Tir$ = 238 K, which represents deep convection conditions with strong downdraft occurrence. Both parameters increase afterward to cleaner sky conditions and out-of-downdraft conditions.

5    In contrast, for $\Delta\theta'_e > +1.5$ at the event's onset, the ensembles show an opposite behavior of the parameters. The $\Delta\theta'_e$ presents an increasing tendency from 10 h before the event until $\Delta t = 0$, reaching a value of +4. Afterward, $\Delta\theta'_e$ decreases, but always retains positive values, indicating that there is a class of growth events that may not be driven by convective downdrafts. The $Tir$ values from 10 h before the events up to $\Delta t = 0$ are equivalent to conditions close to clear skies (Tir $\sim$ 280 K), which agrees with the results obtained for $\Delta\theta'_e$, as an indication of sky conditions not dominated by convection systems around the

10    event's onset. After $\Delta t = 0$, $Tir$ values decrease, but without presenting deep convection conditions. This shows that clear sky



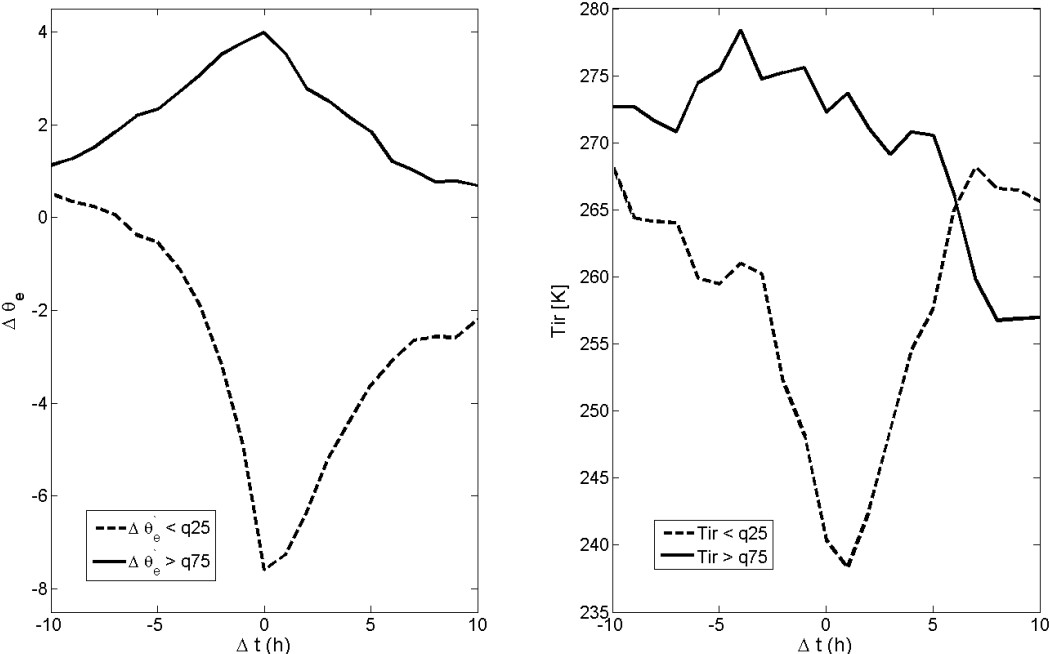

**Figure 13.** Ensemble analysis for a) $\Delta\theta'_e$ and b) $Tir$ considering 10 h before and 10 h after the event's onset. The cases where $\Delta\theta'_e$ at the event's onset is less than the 25th percentile (dashed lines) and more than the 75th percentile (full lines) are shown. In total, 72 cases were considered in the analysis (36 cases for each condition).

events can be associated either with advection or subsidence bringing particulates from another area (e.g., a nearby rain event) or from the upper troposphere.

## 4 Summary and conclusions

This study reports the statistical characterization of aerosol particle growth events in the sub-50 nm size modes (10 - 50 nm), with continuous measurements (February 2014 to September 2020) of PNSDs at a remote site in Central Amazonia. In total, 254 particle growth events were detected, comprising about 14 % of the analyzed days, of which 88 % was found between January to June and 12 % from July to December. The diurnal cycle of the growth events shows that most of them occur during the daytime, accounting for 74%, but with a significant amount of events during the night (26%). In the daytime, 53 % of the growth events start in the early afternoon, between 06:00 and 11:00 00 LT, with a pronounced peak at 07:00 LT, showing a direct relation to the photochemistry and evolution of the PBL. These events also coincide with a precipitation peak in the morning. The nocturnal increase of $N_{<50}$ is likely related to convective systems that result in the precipitation peak in the morning. The subsequent decrease of $N_{<50}$ is likely due to the coagulation process that ages $CN_{<50}$ particles, resulting in growth. A second,





less pronounced but significant peak occurs around 15h LT, coinciding with the strongest precipitation peak, also suggesting a relation to atmospheric convective systems.

The median GR, considering all the growth events, is $5.2\,\mathrm{nm\,h^{-1}}$, which agrees with what was reported by Rizzo et al. (2018). The median $CS_{\mathrm{growth}}$ is $0.0011\,\mathrm{s^{-1}}$. Monthly variations in GR and CS shows that during the wet season the growth events occur under low CS values, although the average $CS_{\mathrm{growth}}$ does not change much from month to month, oscillating around $0.0011\,\mathrm{s^{-1}}$. A remarkable contrast is observed when comparing day (median GR of $5.9\,\mathrm{nm\,h^{-1}}$ and median $CS_{\mathrm{growth}}$ of $0.0012\,\mathrm{s^{-1}}$) and night growth events (median GR of $4.0\,\mathrm{nm\,h^{-1}}$ and median $CS_{\mathrm{growth}}$ of $0.0009\,\mathrm{s^{-1}}$). Daytime events are directly influenced by sunlight, which controls photochemistry and hence the oxidation of SOA precursors. In contrast, nocturnal events may have different causes and mechanisms, which may be related to meteorology, entrainment of air and particles from the free troposphere into the BL, and perhaps the contribution of biogenic sources.

Analysis performed using $\Delta\theta_e'$ and $Tir$ revealed that diverse atmospheric dynamics play different roles during particle growth event days. Many event onsets coincide with downdraft occurrences, when $N_{<50}$ appear and grow afterward. We also observed that accumulation mode particles processed in clouds appear sporadically, causing a prominent Hoppel minimum. The growth events occurring under clear skies present GR and CS higher than those reated to deep convection: the median GR and CS for clear sky conditions are $7.0\,\mathrm{nm\,h^{-1}}$ and $0.0016\,\mathrm{s^{-1}}$, whereas under deep convection conditions, the median GR and CS are $3.8\,\mathrm{nm\,h^{-1}}$ and $0.0005\,\mathrm{s^{-1}}$.

The events were classified according to their frequency of occurrence throughout the day, showing that they are mostly driven by local convective activities (73%). However, when analyzing the growth events by $\Delta\theta_e'$ at the event's onset, the occurrence of downdrafts does not explain all the cases. The contrast is more evident in the ensembles of $\Delta\theta_e'$ and $Tir$ when growth events and their respective occurrence days are selected by $\Delta\theta_e' > 75$th percentile ($\Delta\theta_e' =+1.5$) at the event's onset. For these events, representing about 27% of the growth events, $\Delta\theta_e'$ is maximum and positive at time $\Delta t = 0$, and even ten hours before or after the growth event, it did not present negative values that could indicate the occurrence of convective downdrafts. Also, $Tir$ fluctuates at around 270 K over the observed period, which represents shallow clouds, instead of convective systems conditions. In contrast, events with $\Delta\theta_e' < 25$th percentile ($\Delta\theta_e' =-5.3$) at the event's onset are associated with downdraft occurrences: $\Delta\theta_e'$ strongly decreases 10 hours after the growth event up to $\Delta t = 0$, which is also followed by a strong decrease in $Tir$.

Sources that could explain growth events in the absence of deep convection are likely related to primary biogenic aerosols emitted by the forest, smooth entrainment of air masses from the free troposphere to the PBL in the early morning, and even different meteorological mechanisms such as gravity waves and particle production by lightning in the free troposphere, as reported by Toledo Machado et al. (2021). Another possible explanation is related to nighttime downdrafts far upwind, which get trapped above the nocturnal boundary layer, and travel in the jet above the nocturnal inversion for potentially quite a large distance, being mixed down into the PBL after sunrise, as suggested by Krejci et al. (2005). Therefore future studies are required to unveil the aerosol sources that could explain the diversity of particle growth events observed in the lower troposphere over Central Amazônia.



*Data availability.* The data of the key results presented here have been deposited in supplementary data files for use in follow-up studies. Particle number size distributions for the entire observation period are available via the ATTO data portal through https://www.attoproject.org/. For data requests beyond the available data, please refer to the corresponding authors.

*Author contributions.* MAF and FD contributed equally to this work. MAF, FD, and CP designed the study. MAF and FD analyzed the data.
MAF, FD, LAK, BAH, FGM, JS, SC, JB, and SW collected and processed the ATTO aerosol data. LATM processed the satellte data. AA and MS collected the micrometeorological data at the INSTANT tower at ATTO. SW, FGM, PA, MOA and UP provided essential scientific support for the ATTO measurements. DW supported the data management. MAF, CP, and FD wrote the paper. LATM, MOA, LVR, HMJB, JPN, FGM, MP, SC, JB, and SC contributed with valuable ideas and comments to the analysis and the manuscript. All authors contributed to the discussion of the results as well as the finalization of the manuscript. PA and CP supervised the study.

*Competing interests.* The authors declare that they have no known competing interests that could have influenced the work reported in this paper.

*Disclaimer.* This paper contains results of research conducted under the Technical/Scientific Cooperation Agreement between the National Institute for Amazonian Research, the State University of Amazonas, and the Max-Planck-Gesellschaft e.V.; the opinions expressed are the entire responsibility of the authors and not of the participating institutions.

*Acknowledgements.* This work has been funded by the Max Planck Society (MPG) and FAPESP - Fundação de Amparo à Pesquisa do Estado de São Paulo, grant number 2017/17047-0. MAM acknowledges the financial support of CNPq for the PhD scholarship, project number 169842/2017-7 and CAPES, for a sandwich doctorate at the Max Planck Institute for Chemistry, project number 88887.368025/2019-00. For the operation of the ATTO site, we acknowledge the support by the Max Planck Society (MPG), the German Federal Ministry of Education and Research (BMBF contracts 01LB1001A, 01LK1602B, and 01LK2101B) and the Brazilian Ministério da Ciência, Tec-
nologia e Inovação (MCTI/FINEP contract 01.11.01248.00) as well as the Amazon State University (UEA), FAPEAM, LBA/INPA and SDS/CEUC/RDS-Uatumã. We acknowledge the support by the Instituto Nacional de Pesquisas da Amazônia (INPA). We would like to thank Reiner Ditz, Jürgen Kesselmeier, Susan Trumbore, Alberto Quesada, Thomas Disper, Thomas Klimach, Andrew Crozier, Björn Nillius, Uwe Schulz, Steffen Schmidt, Delano Campos, Sam Jones, Fábio Jorge, Hermes Braga Xavier, Nagib Alberto de Castro Souza, Adir Vasconcelos Brandão, Amauri Rodriguês Perreira, Antonio Huxley Melo Nascimento, Roberta Pereira de Souza, Bruno Takeshi, and Wallace
Rabelo Costa for technical, logistical, and scientific support within the ATTO project.



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
