# Peer review of "Occurrence and growth of sub-50 nm aerosol particles in the Amazonian boundary layer"

_Atmospheric Chemistry and Physics, 2021_

## Referee Comment (RC1)

Based on ~6 years of measurements collected at the ATTO station (Feb. 2014 – Sep. 2020), in the Amazonian forest, the study by Franco et al. investigates the occurrence of particle growth events in the sub-50 nm size range as well as the atmospheric conditions which favour the process. In addition to highlighting an interesting data set, this paper is well written, and I would therefore recommend it for final publication in ACP. Since the growth events have already been reported in different studies, I would however suggest to better highlight what distinguishes this new study from the previous ones, as well as the importance/newness of the results associated with it; indeed, even if it is clearly mentioned that there are gaps in the understanding of the process, some sentences in the paper suggest that a number of elements are already known (e.g. P5, L1-3; P11, L26-29). I also have a series of comments below that I would suggest to address before final publication.

P1, L13: I would recommend (actually throughout the manuscript) using scientific notation to report CS values.

P2, L23: Do the authors mean that the occurrence of growth events in Amazonia is on average less frequent than "classical NPF" at other PBL sites? The wording of the sentence is in my opinion a bit confusing as it gives the impression that it compares the frequency of occurrence, between Amazonia and other sites, of a process that seems characteristic of Amazonia (L21 "Amazonian banana plots", L23 "these characteristic events").

P4, L10-12: If I am not mistaken, among the listed references only Kirby and co-workers report results from laboratory experiments; Rose et al. (2018), in contrast, report complementary observations from "the real atmosphere", and Zhao et al. (2020) uses a combination of measurements and model simulations.

P6, L10-11: "sampling from the 60 m inlet at the 80 m high so-called triangular mast". It is not very clear to me, does it just mean that the particle sampling is not done at the top of the mast but at a lower height?

P6, L20: I would suggest to replace "almost" by "more than" since the data set covers more than 6 years.

P7, L24-25: Can the authors clarify what they mean by "the same fitting routine as for the first (dominant) mode"? Do they mean that once the second (and possibly third) mode size range is identified, the mode diameter identification procedure described in 1. (P7, L18-20) is applied? Also, can the authors say a few words about the criteria for deciding whether a PNSD is best described by 1, 2 or 3 modes? Is this test part of 4. (P7-8, L28-L4)?

P8, L15-16: "moving average with moving windows of 25 minutes for time and 20 nm size for particle diameter". I wonder about the choice of values used for the smoothing algorithm:
- the size range of interest extends over 40 nm, between 10 and 50 nm (it is in fact restricted to 10-40 nm for the application of the procedure from Kulmala et al., 2012): is the 20 nm window used for the smoothing average not too wide, and does not impact the monitoring of the process evolution through this relatively restricted size range?
- similarly, is the 25 min window not too large for the description of a phenomenon that (may) have a relatively sudden character?

P9, L15: How was the threshold on $R^2$ selected (> 0.6)?

P9, Meteorological parameters: Outside the period January 2019 - September 2020, the measurements of the different meteorological variables involved in the calculation of the equivalent

potential temperature (i.e. T, p and RH) were not made at the same height (55, 55 and 81 m, respectively). On the other hand, between January 2019 and September 2020, the measurements of these same variables were made at a significantly higher height (321 m; especially compared to the measurement height of the SMPS). Doesn't this have an impact on the analysis? In particular, I wonder about the possible existence, in the vicinity of the forest canopy, of a very fine scale tropospheric stratification phenomenon, like the one observed by Zha et al. (2018) over the boreal forest?

P11, L19-20, 23-24: Since the diameters that are reported are for average distributions, I would suggest clearly indicating "on average", or "(mean)", same as for the description of Fig. 5.

P12, L15: In light of the results of this work, which suggest that there are growth events that may not be directly related to characteristic wet season conditions such as rain / deep convection events, I would suggest changing the sentence slightly, replacing "indicating" by "suggesting". Related to my general comment, this would contribute to give more weight to the results of the present study.

P13, L2: Isn't the entire measurement period between February (and not April) 2014 and September 2020?

P13, study of $CS_{growth}$: As proposed, the definition and analysis of $CS_{growth}$ does describe the decrease in CS related to a decrease in the concentration of > 50-100 nm particles visible at the event onset, but $CS_{growth}$ also includes the contribution of the event itself to CS; therefore, $CS_{growth}$ is likely ultimately impacted by the strength of the event. Based on Fig. 10, it seems to be the case in particular during "deep convection events", and it is more broadly suggested by the similar "trends "obtained for $CS_{growth}$ and GR , which both seem to represent a measure of the strength of the event. If the objective is to study the conditions that favor the occurrence of events, why not look (at least in addition to the analysis of $CS_{growth}$) at the evolution of the CS related only to particles >50 or even 100 nm, i.e. associated to Aitken / accumulation mode particles?

P17-20, Diurnal trends:
- Why did the authors not also present for all the variables shown in Fig. 7, in the same way as for $N_{50}$, the median daily variation observed on event days? Even if the number of event days is limited (and the corresponding statistics must therefore be considered with caution) this might help to illustrate what distinguishes in particular these days from non-event days during the wet season.
- P18, L11-12: "This suggests that $CN_{<50}$ are injected into the PBL by rainfall events during the late afternoon and early night and last until mid-morning": beyond lasting until mid-morning, the particles assumed to have been injected in the late afternoon and early evening show an increase in concentration (already between 15:00 and 00:00 LT but more importantly between 00:00 and 09:00 LT). Can the authors comment on this increase? Is it related to the dynamics of the PBL described in L25-31? If so, the link should be more clearly established. Also, while the decrease of $N_{<50}$ is mentioned in the Summary and conclusions, it is not discussed in Sect. 3.4, whereas I think this would be interesting for a more complete description in the results section dedicated to the analysis of diurnal cycles. Finally, still concerning the $N_{<50}$ analysis, I would not say that the concentrations observed on event days are *significantly* higher. The difference on the medians is most pronounced between 04:00 and 14:00 LT but it does not exceed ~35 $cm^{-3}$ (corresponding to a multiplying factor of 1.6 compared to all data), and it is less than 20 $cm^{-3}$ on the rest of the day. Considering that concentrations are moreover likely affected by uncertainties, I would suggest to slightly balance this observation.

- With the exception of $P_{ATTO}$, the analysis of the different meteorological variables shown in Figure 7 is relatively brief, and I find it overall decoupled from the analysis of $N_{50}$ and growth events (I find that the link between the different observations is better established in the conclusion section!). For example, if the occurrence of the growth events seems to be often connected to a rainy episode, there seems, in addition, to be a strong link between the onset of the growth process and radiation; this should for instance be highlighted as a support for "daytime events are directly influenced by sunlight".
- This leads me to a broader question: the conditions that favor the occurrence of a growth event are implicitly favorable, in the first place, to the "appearance" of <50 nm particles. However, are there times when the appearance of <50 nm particles is not followed by the growth of these particles? Such days would allow the identification of conditions that are favourable in particular to the growth phase if this is not systematic.
- P18, L9: I assume that the numbers in brackets correspond to quartiles; this should be specified at first use.

P19, L3 - P20, L1-5: With the exception of light, which is by definition specific to daytime events, the list of factors mentioned to explain the difference between daytime and nighttime events is not clear to me. In particular, as further illustrated by the analysis of the different event groups in Sect. 3.5, there is a likely role of weather (and in particular convection/downdraft) in the case of groups G1-G3, i.e. including both daytime and nocturnal events; moreover, the entrainment of particles from the upper troposphere into the PBL is mentioned for nocturnal events, but it is this mechanism that is, in one way or another, at the origin of the transport of particles also for daytime events, isn't it? And isn't the contribution of biogenic sources mentioned for the nocturnal events also valid for the daytime?

P23: For the sake of consistency in the abbreviations used, I would suggest changing "BL" to "PBL".

P23, L14: To avoid any confusion, I would suggest to change Di to another acronym, as Di is already used in Sect. 2.4 in the description of the fitting process (where it corresponds to the diameter of mode i).

P23, L15: "The results show a clear difference": I would suggest to slightly balance this assertion with respect to Tir as, in a first approach, all groups (with the exception of G3 maybe) have relatively comparable median Tir, at least all belonging to the same range corresponding to mixed sky conditions.

P24, L7-8: When they speak of a stable Di, I assume that the authors mean that Di varies little from one event to another; if this is indeed the case, I would suggest not using the expression "remains stable", which, in my opinion, gives the impression of referring to a temporal evolution of the diameter. I would for example suggest "while the events belonging to G1 and G4 display comparable median Di".

P24, L14: Change "15 LT" to "15:00 LT" for consistency in the notation of times.

P24, L20: The value reported for G3 median Di in the text (24.3 nm) is slightly different from that reported in Table 2 (24.6 nm).

Fig. 3: Change "a particle growth event" to "2 particle growth events" in the figure caption.
Regarding the color bar, I would suggest indicating the concentrations in logarithmic scale instead of showing the logarithm of the concentrations, to facilitate the interpretation of the figure. If the authors

wish to keep this representation, however, the unit should be corrected (log10(particle concentration) is not in cm$^{-3}$). The same comment applies to similar figures.

Fig. 7: Ticks on the x-axis are not located at the same place in all panels, which makes it difficult to read the times from the lowest panel.

Fig. S3: abbreviation "abr" for April should be changed to "apr".

References:

Zha, Q., Yan, C., Junninen, H., Riva, M., Sarnela, N., Aalto, J., Quéléver, L., Schallhart, S., Dada, L., Heikkinen, L., Peräkylä, O., Zou, J., Rose, C., Wang, Y., Mammarella, I., Katul, G., Vesala, T., Worsnop, D. R., Kulmala, M., Petäjä, T., Bianchi, F., and Ehn, M.: Vertical characterization of highly oxygenated molecules (HOMs) below and above a boreal forest canopy, Atmos. Chem. Phys., 18, 17437–17450, https://doi.org/10.5194/acp-18-17437-2018, 2018.

---

## Author Comment (AC1)

**Responses to the review of the manuscript "Occurrence and growth of sub-50 nm aerosol particles in the Amazonian boundary layer", by Franco et al., submitted for publication in ACP - manuscript ACPD 2021-765**

Dear editor, we would like to thank you and both reviewers for their valuable comments and useful suggestions to improve our manuscript. Below you can find answers and actions for each individual comment from the reviewers. In order to make it easier to identify the individual answers and actions, we used the following color code strategy:

- **In black, the referee's comments.**
- **In blue, the author's responses.**
- ***In blue and italic, the text modifications we made in the manuscript.***

**Responses to Anonymous Referee #1**

Based on ~6 years of measurements collected at the ATTO station (Feb. 2014 – Sep. 2020), in the Amazonian forest, the study by Franco et al. investigates the occurrence of particle growth events in the sub-50 nm size range as well as the atmospheric conditions which favour the process. In addition to highlighting an interesting data set, this paper is well written, and I would therefore recommend it for final publication in ACP. Since the growth events have already been reported in different studies, I would however suggest to better highlight what distinguishes this new study from the previous ones, as well as the importance/newness of the results associated with it; indeed, even if it is clearly mentioned that there are gaps in the understanding of the process, some sentences in the paper suggest that a number of elements are already known (e.g. P5, L1-3; P11, L26-29). I also have a series of comments below that I would suggest to address before final publication.

We thank referee #1 for the constructive comments and useful suggestions. They helped us to clarify important aspects of our analysis and, thus, to improve the manuscript overall. We agree with the referee that the manuscript would profit from clearer statements on what differentiates this study from others previously published. We extended the discussions on the main scientific questions we are addressing, clarifying the arguments on why we have done this study. The ideas are based on questions we have for at least 20 years of aerosol size distribution measurements in Amazonia. This was added in the last paragraph when we finished the review of the current literature and introduce the novelty of this study.

*This study aims to identify and characterize the occurrence of particle growth events in the size range from 10 to 50 nm within the PBL of Central Amazonia. While previous studies have documented the occurrence and properties of freshly nucleated particles at high altitudes (Krejci et al., 2003; Andreae et al., 2018; Williamson et al., 2019), their growth in the course of downward transport (Wang et al., 2016a), as well as the appearance of sub-50 nm particles in the PBL (Rizzo et al., 2018; Wimmer et al., 2018), major questions remain open. With this study, we take a step beyond the existing knowledge, based on more than 6 years of aerosol measurements and complementary meteorological and satellite observations. In particular, we focus on a statistically broad characterization of Amazonian particle growth events ('Amazonian bananas') by means of GR, CS, seasonality, diurnal cycle, as well as their relationship to meteorological variables and deep convection. We also document growth events under clear sky conditions and thus in the absence of deep convective mixing. Therefore, the knowledge obtained here about the sub-50 nm particle growth events addresses an important gap in our understanding of the Amazonian aerosol life cycle and will help to constrain the CCN sources and properties in this globally important ecosystem.*

Please find below our answers to each of the referee's comments, suggestions and questions.

P1, L13: I would recommend (actually throughout the manuscript) using scientific notation to report CS values.

We revised the manuscript and implemented the scientific notation to report CS values throughout the text.

P2, L23: Do the authors mean that the occurrence of growth events in Amazonia is on average less frequent than "classical NPF" at other PBL sites? The wording of the sentence is in my opinion a bit confusing as it gives the impression that it compares the frequency of occurrence, between Amazonia and other sites, of a process that seems characteristic of Amazonia (L21 "Amazonian banana plots", L23 "these characteristic events").

Thanks for pointing that out. It was not clear in the original manuscript that there are two different aspects of NPF events in Amazonia compared to boreal forests: the size range covered by these events as well as the frequency of occurrence. One clear difference is that Amazonian NPF events at the PBL are observed starting at about 20-40 nm. In boreal forest sites, such as Hyytiälä, they are documented to happen to start at much smaller sizes (3-10 nm). They are also much more frequent in Hyytiälä than in Amazonia (Rizzo et al., 2018). In order to improve the clarity of this important point, we modified the whole paragraph, which reads now as follows:

*In the long list of locations where the 'classical NPF' has been detected in the planetary boundary layer (PBL) (Kerminen et al., 2018), the Amazon rain forest is a remarkable exception (e.g., Andreae, 2013; Rizzo et al., 2018; Wimmer et al., 2018). Here, events have been observed that indeed resemble the classical 'banana plots' of NPF but differ clearly in the initial diameter of the growth curve. While the smallest diameters in, for instance, boreal forest is typically in the few nm range, the 'Amazonian bananas' rather start between about 20 to 40 nm (Kulmala et al., 2012; Kerminen et al., 2018). Rizzo et al. (2018) discussed the occurrence of such sub-50 nm particle growth events in the Amazon and found them only in 3% of the 749 days examined, associated mainly with convective downdrafts. Accordingly, the 'Amazonian bananas' start at larger diameters and are comparatively rare relative to the classical events e.g., in boreal forests (Nieminen et al., 2018; Dada et al., 2018).*

P4, L10-12: If I am not mistaken, among the listed references only Kirby and co-workers report results from laboratory experiments; Rose et al. (2018), in contrast, report complementary observations from "the real atmosphere", and Zhao et al. (2020) uses a combination of measurements and model simulations.

That is correct. Thus, we corrected the references and the fragment in the revised manuscript, which now reads as:

*Recent evidence of pure biogenic ion-induced nucleation under controlled laboratory (Kirkby et al., 2016) and under real atmospheric conditions (Rose et al., 2018; Zhao et al., 2020) highlights possible mechanisms for NPF pathways in a clean atmosphere.*

P6, L10-11: "sampling from the 60 m inlet at the 80 m high so-called triangular mast". It is not very clear to me, does it just mean that the particle sampling is not done at the top of the mast but at a lower height?

The tower where the measurements were conducted is 80 m high, but the sampling was performed at a height of 60 m because the inlet line is 60 m long. In order to improve clarity, the text now reads as:

*This study focuses on particle number size distributions (PNSDs) obtained from a Scanning Mobility Particle Sizer (SMPS) with an inlet located at 60m above ground. The inlet used to sample the aerosols is installed on an 80m high tower (02° 08.602'S, 59° 00.033'W; 130m a.s.l.) at the ATTO site. The SMPS is manufactured by TSI Inc., and we used as classifiers: model 3080 and, later,*

*model 3082, coupled to a condensation particle counter (CPC) 3772. The inlet height was chosen to be approximately 30m above the average canopy height, which enables measurements close to the canopy without direct contact with the largest trees. The SMPS is located in an air-conditioned laboratory container at the foot of the mast. Sample air is transported through a 25mm diameter stainless steel tube (finetron tubes, Dockweiler AG, Neustadt-Glewe, Germany) and dried to a relative humidity (RH) below 40 %.*

P6, L20: I would suggest to replace "almost" by "more than" since the data set covers more than 6 years.

*Thanks, we replaced the word as suggested, and the sentence now reads as:*

*The SMPS measurements cover the particle size range from 10 to 400 nm and yield a temporal resolution of 5 min. The PNSD data covers more than six years, from February 2014 to September 2020, covering 1 596 measurement days, and comprising 426 272 sample runs in total. The data coverage of 67 % over the entire time frame (i.e., Feb 2014 to Sep 2020) can be considered as a robust data foundation and statistical basis for the observations and conclusion presented here.*

P7, L24-25: Can the authors clarify what they mean by "the same fitting routine as for the first (dominant) mode"? Do they mean that once the second (and possibly third) mode size range is identified, the mode diameter identification procedure described in 1. (P7, L18-20) is applied? Also, can the authors say a few words about the criteria for deciding whether a PNSD is best described by 1, 2 or 3 modes? Is this test part of 4. (P7-8, L28-L4)?

*The term "the same fitting routine" refers to step 1 in which the initial best Dp is determined. The algorithm fits three modes in their respective size ranges, and a bin-wise comparison between the fit and the SMPS measurement is performed to check the quality of the resulting fit based on statistics. To clarify the procedure, we removed the "(if present)". In order to clarify the methodology, and we improved the text in the manuscript, which now reads as:*

*1. In the first step, the maximum particle number concentration and the corresponding particle diameter, $D_{dom}$, are determined within the particle number size distribution. Within the size range of -30 % to +20 % of $D_{dom}$, a one-modal log-normal distribution is fitted.*

*2. The first one-modal fit is assigned as accumulation ($D_{Acc} \in [100,300]$), Aitken ($D_{Ait} \in [50,100)$) or sub-50 nm mode( $D_{<50} \in [9,50)$), and two additional one-modal log-normal distributions are added for the remaining modes. The parameters of the three log-normal distributions are then varied within the mentioned diameter range, the standard deviation, and for concentrations less than the maximum of the particle number size distribution.*

*3. The geometric standard (i) deviation of all modes was constrained within the range of 1.1 to 1.55, which was optimized for the ATTO conditions.*

*4. Subsequently, a joint optimization of the previously obtained fit parameters ($D_i$, $\sigma_i$, and $N_i$) for the modes was conducted. The procedure is developed by fixing two of the modes and leaving the third free so that its parameters are again optimized by minimizing the least-squares. The optimization order in this process was to optimize the sub-50 nm mode, then the accumulation mode, and, finally, the Aitken mode. In this case, all the free diameters of the modes could vary between 0.5 Di and 1.5 Di. As a measure of fitting quality, for each particle number size distribution, the algorithm compares the particle number concentrations of each bin of the measured and the fitted curve and obtains the $R^2$ value. We considered only fits in which the agreement returned $R^2 > 0.8$, which means that about 97% of the data are covered by the developed mode fitting. Examples of fits can be seen in Figure S1. 20*

*5. Comparisons between the integrated particle number concentration from the SMPS*

*measurements ($N_{conc,SMPS}$) and lognormal fitted size distributions ($N_{conc,\sum n-modes}$) were made to further assure the quality of the fits. Within this data set, on average, fits with $R^2 = 0.97$ were obtained, which yielded a linear fit of $N_{conc,SMPS}$ and $N_{conc,\sum n-modes}$ with $R^2 = 0.99$ (Figure S2).*

P8, L15-16: "moving average with moving windows of 25 minutes for time and 20 nm size for particle diameter". I wonder about the choice of values used for the smoothing algorithm:
- the size range of interest extends over 40 nm, between 10 and 50 nm (it is in fact restricted to 10-40 nm for the application of the procedure from Kulmala et al., 2012): is the 20 nm window used for the smoothing average not too wide, and does not impact the monitoring of the process evolution through this relatively restricted size range?
- similarly, is the 25 min window not too large for the description of a phenomenon that (may) have a relatively sudden character?
-

Thanks for pointing that out. Indeed, in the way it was written, the description is not precise. The moving window has two dimensions: one in time. SMPS measurements last 5 minutes to get a full-size distribution. In order to reduce noise, we average the time window at 25 minutes. A second parameter is the particle size, which actually is not 20 nm as written by mistake before. The correct size window is 5 SMPS--bins. This is also made to reduce noise events in terms of particle size. These two choices were shown as ideal to get reliable data, following suggestions by Kulmala et al., 2012. We also performed several tests to verify if sudden events were missing, and, in general, the method could cover them without significant losses. Therefore, the text was improved as follows:

*All data were smoothed to eliminate single exceptionally high or low values to avoid possible bias due to short intense particle peaks or dips. The moving window has two dimensions: one in time and the other in size. The SMPS measurements last 5 minutes to get a full-size distribution. In order to reduce noise, we average the time window at 25 minutes. The second parameter is the particle size window, which accounts for 5 SMPS-bins. This is also made to reduce noise in terms of particle size. These two choices were shown as ideal to get reliable data, following suggestions by Kulmala et al. (2012). We also performed several tests to verify if sudden events were missing, and found that, in general, the method could cover them without significant losses.*

P9, L15: How was the threshold on $R^2$ selected (> 0.6)?

Several tests were carried out varying the threshold value of $R^2$ to obtain the most suitable fits, but which could also represent more subtle growth events, sometimes difficult to detect. Thus, we found that the fits with $R^2 = 0.6$, with p-value < 0.05, were able to represent the widest possible variability of events, without compromising the analyses. In order to make the selection process clearer in the manuscript, we modified the text, which now reads as:

*To assure the data quality during the analyses, fits were statistically tested and only fits with $R^2 > 0.6$ and p-value < 0.05 were accepted. Additionally, we performed visual inspections of the quality of each of the fits. It is worth mentioning that fits statistically tested with $R^2 > 0.6$ were able to represent the widest possible variability of growth events, without compromising the analyses.*

P9, Meteorological parameters: Outside the period January 2019 - September 2020, the measurements of the different meteorological variables involved in the calculation of the equivalent potential temperature (i.e. T, p and RH) were not made at the same height (55, 55 and 81 m, respectively). On the other hand, between January 2019 and September 2020, the measurements of these same variables were made at a significantly higher height (321 m; especially compared to the measurement height of the SMPS). Doesn't this have an impact on the analysis? In particular, I wonder about the possible existence, in the vicinity of the forest canopy, of a very fine scale tropospheric stratification phenomenon, like the one observed by Zha et al. (2018) over the boreal forest?

Unfortunately, the meteorological data was not available at the 80 m tower for the final period of analysis. Thus, for 2019 and 2020, we had to use data from the 325 m tower. As pointed out by the reviewer, the important issue is whether measurements at the two heights would impact the values of $\Delta\theta_{e'}$. To test this, we used two months of simultaneous measurements at the two levels: ~60 and 325 m, from January to February 2019. The $\Delta\theta_{e'}$ time series for both months are shown in Figure 1.

[Figure]

Figure 1: time series of $\Delta\theta_{e'}$ calculated with meteorological measurements obtained simultaneously at ~60 and 325m high.

The correlation plot of $\Delta\theta_{e'}$ calculated for these two months is displayed in Figure 2, and the regression equation obtained with a linear fit is $\Delta\theta_{e',325m} = \Delta\theta_{e',60m} \times 0.97 - 0.12$, with p-value < 0.05 and R² = 0.75. In addition, $\Delta\theta_{e'}$ had the same sign at both altitudes, showing that the detection of downdrafts was not affected by the use of data from the height of 325 m. Overall, the result shows that the use of meteorological data from 325 m instead of 60 m does not impact significantly the values used for $\Delta\theta_{e'}$, and therefore, does not impact the conclusions of the analysis.

We can not rule out a very fine scale of tropospheric stratification phenomenon close to the canopy, as observed by Zha et al. (2018). However, the tower is situated over a quite flat area, with a canopy height homogeneous over several kilometers. This would be a good future study for the ATTO site.

[Figure]

Figure 2: correlation plot of $\Delta\theta_{e'}$ calculated with meteorological data obtained at ~60 and 325 m high. The blue line is the linear fit obtained from the data adjust.

We added a paragraph on page 11, Lines 6 - 16 of the revised version clarifying this issue. It reads as:

*It should be mentioned that two time series of meteorological data were used to calculate $\Delta\theta_{e'}$: the first one, with measurements conducted close to the canopy (2013 - 2018), and the second one, with measurements conducted at 325 m elevation (2019 - Sep 2020). This was necessary because meteorological data are not available at the 80 m tower for the final period of analysis. The consistency of the $\Delta\theta_{e'}$ calculation was verified by comparing $\Delta\theta_{e'}$ for a 2-month period with overlapping measurements at the two height levels (January and February 2019), as shown in Figure S4. Figure S5 shows the correlation between $\Delta\theta_{e'}$ obtained at the two levels, with the statistical results of the comparison. Although there are very small differences for single pairs of measurements, the overall agreement is reasonably good, and therefore, does not impact the conclusions of the analysis. These results encouraged us to use the meteorological data measured at 325 m height, which enabled us to extend the data analysis to the years 2019 and 2020. We can not rule out a very fine scale stratification phenomenon close to the canopy, as observed by Zha et al. (2018), which could have some influence on $\Delta\theta_{e'}$ but to a minor extent, as observed by the comparison analysis. Further studies are required to examine this aspect in detail for the ATTO site.*

P11, L19-20, 23-24: Since the diameters that are reported are for average distributions, I would suggest clearly indicating "on average", or "(mean)", same as for the description of Fig. 5.

We incorporated the suggestions and it now reads as (P12, L12-16):

*On average, the Aitken mode is centered at 71 nm, the Hoppel minimum is centered at 102 nm, and the accumulation mode is centered at 153 nm. In contrast, Figure 2b shows the typical dry season PNSDs characterized by a strong mono-modal shape with a dominating accumulation mode, reflecting the prevalence of biomass burning pollution (e.g., Rissler et al., 2006; Brito et al., 2014). On average, the accumulation mode is centered at 146 nm.*

P12, L15: In light of the results of this work, which suggest that there are growth events that may not be directly related to characteristic wet season conditions such as rain / deep convection events, I would suggest changing the sentence slightly, replacing "indicating" by "suggesting". Related to my general comment, this would contribute to give more weight to the results of the present study.

Thanks for the suggestion. Indeed it gives much more weight to the results. We incorporated it, and the new sentence now reads as:

*Note that in all previous studies in Amazonia, the growth events were observed during the wet season, suggesting that this event type is a typical wet season phenomenon associated with precipitation (Zhou, 2002; Wimmer et al., 2018; Rizzo et al., 2018).*

P13, L2: Isn't the entire measurement period between February (and not April) 2014 and September 2020?

Indeed the measurements considered in the analysis started in February 2014. The sentence now reads as follows:

*For the entire measurement period (February 2014 – September 2020) this corresponds to about 30 event days per year.*

P13, study of $CS_{growth}$: As proposed, the definition and analysis of $CS_{growth}$ does describe the decrease

in CS related to a decrease in the concentration of > 50-100 nm particles visible at the event onset, but $CS_{growth}$ also includes the contribution of the event itself to CS; therefore, $CS_{growth}$ is likely ultimately impacted by the strength of the event. Based on Fig. 10, it seems to be the case in particular during "deep convection events", and it is more broadly suggested by the similar "trends "obtained for $CS_{growth}$ and GR , which both seem to represent a measure of the strength of the event. If the objective is to study the conditions that favor the occurrence of events, why not look (at least in addition to the analysis of $CS_{growth}$) at the evolution of the CS related only to particles >50 or even 100 nm, i.e. associated to Aitken / accumulation mode particles?

Yes, the condensational sink (CS) is influenced by the smaller particles as well, since it is calculated based on the entire size range. Nevertheless, the CS is proportional to the particle surface area and, hence, anyhow more sensitive to changes in the accumulation mode. We do not expect significantly different results by calculating CS for a selected size range and prefer to stick to the established definition taking into account the entire size spectrum.

P17-20, Diurnal trends:

- Why did the authors not also present for all the variables shown in Fig. 7, in the same way as for $N_{50}$, the median daily variation observed on event days? Even if the number of event days is limited (and the corresponding statistics must therefore be considered with caution) this might help to illustrate what distinguishes in particular these days from non-event days during the wet season.

We did not discriminate meteorological variables with respect to days with and without events, because during the analyses, we did not observe significant differences between the days of events and non-events. Furthermore, in comparison with the wet season means, no meteorological variable showed significant differences either. In fact, only the $N_{<50}$ component showed significant differences, so we decided to show it compared to the others within the diurnal cycle. Since the vast majority of aerosol growth events occur in the wet season months, we decided to present the contrast of meteorological variables in relation to the entire period and only in the wet season. To clarify this aspect, we improved a fragment of the text of the revised manuscript, which now reads as:

*The meteorological variables, T, SW, RH, and visibility were not discriminated with respect to days with and without events, because no significant differences between event and non-event days were observed. Possible effects of deep convection, associated rainfall, and cloudiness are investigated in the following Sec. 3.5.*

- P18, L11-12: "This suggests that $CN_{<50}$ are injected into the PBL by rainfall events during the late afternoon and early night and last until mid-morning": beyond lasting until mid-morning, the particles assumed to have been injected in the late afternoon and early evening show an increase in concentration (already between 15:00 and 00:00 LT but more importantly between 00:00 and 09:00 LT). Can the authors comment on this increase? Is it related to the dynamics of the PBL described in L25-31? If so, the link should be more clearly established.

The increase in $CN_{<50}$ is likely due to the dynamics of the PBL and the particle transport from the free troposphere into the PBL through downdrafts (rainfall events) and, likely, entrainment too. The study from Machado et al., 2021 discusses and illustrates this process in more detail, although it is worth mentioning that there are still open questions regarding the mechanisms and sources related to this nocturnal $CN_{<50}$ increase. To clarify and effectively establish a link with the dynamics described in L25-31, as suggested by the reviewer, we improved the text, which now reads as:

*The evolution of the PBL also has a strong influence on the diurnal pattern. At night, the nocturnal PBL close to the forest canopy is decoupled from the residual layer above (Fisch et al., 2004). In the morning hours – as soon as convection becomes effective – air masses transported into and within the residual layer are mixed into lower levels and measured at the canopy level. Consequently,*

*CN<50 and Aitken mode particles advected with the residual layer will be mixed downwards and appear at the 60m inlet in the morning hours, typically around 8:00 LT. This behavior is in agreement with that observed in Figure 7e, with the increase of $CN_{<50}$ throughout the night and in the early morning. Machado et al. (2021) discuss this daily mechanism of particle growth in more detail. Section 3.5 further discusses the meteorological conditions regarding convective downdrafts and the atmospheric conditions under which the growth events are observed.*

- Also, while the decrease of N<50 is mentioned in the Summary and conclusions, it is not discussed in Sect. 3.4, whereas I think this would be interesting for a more complete description in the results section dedicated to the analysis of diurnal cycles.

We agree with the reviewer that Sect. 3.4 lacks details about the decrease in $N_{<50}$. Thus, we improved the paragraph of P20, l 9-13 of the revised manuscript, including now a discussion about the decrease of $N_{<50}$, as suggested by the reviewer. The text now reads as follows:

*The diurnal cycle of the growth event onsets has a rather broad maximum spanning over the early morning hours, from 06:00 to 10:00 LT. It peaks at about 07:00 LT and then gradually decreases towards noon (see Fig. 7f), which is in agreement with what is observed in Figure 7e. The morning peak in the occurrence of growth events is likely the main factor related to the decrease in $N_{<50}$, shown in Figure 7e. Processes like condensation and (to a minor extend) coagulation age the $CN_{<50}$ particles, which results in growth. It is interesting to note that the RH maximum shown in Figure 7c coincides with the minimum in FOG occurrence (Figure 7d) and the morning growth event peaks, which indicates that condensational processes indeed play an important role at this period.*

*The diurnal cycle of the growth event onsets has a rather broad maximum in the early morning hours from 06:00 to 10:00 LT. It peaks at about 07:00 LT and then gradually decreases towards noon (see Fig. 7f), which is in agreement with what is observed in Figure 7e. In addition to the PBL development, particles are subject to atmospheric aging and likely condensation of semi and low-volatile compounds resulting in particle growth and a decrease in particle number concentration. It is interesting to note that the morning growth event maximum coincides with a maximum in RH and the occurrence of fog (cf. Fig. 7c,d). A second local and less pronounced maximum is visible from 13:00 to 15:00 LT. The growth events reported during daytime likely correspond to rainfall events as reported by Machado et al. (2021) and probably the vertical transport of $CN_{<50}$ and Aitken size particles due to strong downdrafts in the course of convective rainfall, and the injection of these particle populations into the PBL as reported in Wang et al. (2016a) and Andreae et al. (2018).*

- Finally, still concerning the N<50 analysis, I would not say that the concentrations observed on event days are *significantly* higher. The difference on the medians is most pronounced between 04:00 and 14:00 LT but it does not exceed ~35 cm-3 (corresponding to a multiplying factor of 1.6 compared to all data), and it is less than 20 cm-3 on the rest of the day. Considering that concentrations are moreover likely affected by uncertainties, I would suggest to slightly balance this observation.

We agree with the reviewer and we replaced the term "significantly higher" to "relatively higher", and modified the text to balance the observation, as suggested by the reviewer, which now reads as:

*The particle concentration on growth event days is somewhat higher than that including all analyzed PNSDs, with median daily values and interquartile range of 64 (38 - 108) $cm^{-3}$, compared to 49 (29 - 81) $cm^{-3}$ for all days.*

- With the exception of P_ATTO, the analysis of the different meteorological variables shown in

Figure 7 is relatively brief, and I find it overall decoupled from the analysis of N$_{50}$ and growth events (I find that the link between the different observations is better established in the conclusion section!). For example, if the occurrence of the growth events seems to be often connected to a rainy episode, there seems, in addition, to be a strong link between the onset of the growth process and radiation; this should for instance be highlighted as a support for "daytime events are directly influenced by sunlight".

We agree with the reviewer that the discussions of some meteorological variables are relatively brief. Thus, we improved the discussion regarding the decrease in $N_{<50}$ linking it with the maximum in RH and minimum in FOG occurrence, which now reads as follows:

*The diurnal cycle of the growth event onsets has a rather broad maximum in the early morning hours from 06:00 to 10:00 LT. It peaks at about 07:00 LT and then gradually decreases towards noon (see Fig. 7f), which is in agreement with what is observed in Figure 7e. In addition to the PBL development, particles are subject to atmospheric aging and likely condensation of semi and low-volatile compounds resulting in particle growth and a decrease in particle number concentration. It is interesting to note that the morning growth event maximum coincides with a maximum in RH and the occurrence of fog (cf. Fig. 7c,d).*

In order to highlight the role of solar radiation in the growth process and make clear the links between them (also exploring the information of SW - Figure 7a), we improved the revised manuscript, which now reads as follows:

*Daytime events are directly influenced by sunlight, which controls photochemistry and hence the oxidation of SOA precursors. In contrast, nocturnal events may have different causes and mechanisms, which may be related to meteorology, entrainment of air and particles from the free troposphere into the BL, and perhaps the contribution of biogenic sources.*

- This leads me to a broader question: the conditions that favor the occurrence of a growth event are implicitly favorable, in the first place, to the "appearance" of <50 nm particles. However, are there times when the appearance of <50 nm particles is not followed by the growth of these particles? Such days would allow the identification of conditions that are favourable in particular to the growth phase if this is not systematic.

Yes, there are days when $CN_{<50}$ appears, but there are no clear signs of particle growth. We tried to discriminate these events and look for some relationship with meteorological variables, Tir and $\Delta\theta_{e}$'', but we could not find any clear relationship for the occurrence of particle growths. The only variable that showed any indication for the occurrence of growth was CS. The growth events occurred in periods with relatively smaller CS than those without such events (for example, see Figure 6d) and e), where the peak of growth events occurs with the lowest CS value). This indicates that cleaner atmospheres are more prone to aerosol growth events. Future studies are required to seek relationships that can explain in greater detail the mechanisms that favor particle growth events in Central Amazonia.

- P18, L9: I assume that the numbers in brackets correspond to quartiles; this should be specified at first use.
  Thanks for the suggestion. We improved the text, which now reads as follows:

*The particle concentration on growth event days is somewhat higher than that including all analyzed PNSDs, with median daily values and interquartile range of 64 (38 - 108) $cm^{-3}$, compared to 49 (29 - 81) $cm^{-3}$ for all days.*

P19, L3 - P20, L1-5: With the exception of light, which is by definition specific to daytime events, the list of factors mentioned to explain the difference between daytime and nighttime events is not clear to me. In particular, as further illustrated by the analysis of the different event groups in Sect. 3.5, there is a likely role of weather (and in particular convection/downdraft) in the case of groups G1-G3, i.e. including both daytime and nocturnal events; moreover, the entrainment of particles from the upper troposphere into the PBL is mentioned for nocturnal events, but it is this mechanism that is, in one way or another, at the origin of the transport of particles also for daytime events, isn't it? And isn't the contribution of biogenic sources mentioned for the nocturnal events also valid for the daytime?

There are two general types of aerosol sources not related to long-range transport that contribute to the aerosol population in Central Amazonia: primary biogenic emissions and SOA formed in the UT. We have good evidence that the UT source exists and plays an important role for both daytime and nighttime events. On the other hand, while there is evidence for the emission of primary biogenic particles, (Pöhlker et al., 2012, Moran-Zuolaga et al., 2018), the exact source mechanisms and their role in aerosol particle production are much more speculative. In particular, nighttime growth could be supported both by continued deposition of condensables formed during the day and by production of condensables at night by ozonolysis reactions. For both sources (if both are relevant), fundamental open questions exist about how this particle formation depends on meteorology and aerosol loading in the atmosphere, for example. For UT-NPF, upcoming flight campaigns could likely provide new important insights on the main drivers of the sources for the different particle growth events. It is worth noting that, for biogenic emissions, there is still the need to identify the particle sources, although earlier studies have shown that the growth of secondary aerosol particles can be initiated by biogenically emitted potassium-salt–rich particles (Pöhlker et al., 2012). Also, a primary source close in the canopy would be fundamental for the aerosol particle maintenance in the PBL (Rizzo et al., 2018).

In order to make it clear in the text, we restructured the paragraph mentioned and improved the Summarry and Conclusions section including the following paragraph:

*Daytime events are directly influenced by sunlight, which controls photochemistry and hence the oxidation of SOA precursors. In contrast, nocturnal events may have different causes and mechanisms. One particular mechanism for nighttime growth events could be supported both by continued deposition of condensables formed during the day and by production of condensables at night by ozonolysis reactions. However, the direct influence of meteorology, entrainment of air masses, and perhaps the contribution of biogenic sources can not be ruled out. Upcoming flight and in-situ campaigns are expected to provide new important insights on the main drivers of the sources for the different particle growth events. It is worth noting that, for primary biogenic emissions, there is still the need to identify the particle sources, although earlier studies have shown that the growth of secondary aerosol particles can be initiated by biogenically emitted potassium-salt–rich particles (Pöhlker et al., 2012). Also, a primary source close in the canopy would be fundamental for the aerosol particle maintenance in the PBL (Rizzo et al., 2018).*

P23: For the sake of consistency in the abbreviations used, I would suggest changing "BL" to "PBL".

Many thanks, we replaced BL with PLB throughout the text, as suggested.

P23, L14: To avoid any confusion, I would suggest to change Di to another acronym, as Di is already used in Sect. 2.4 in the description of the fitting process (where it corresponds to the diameter of mode i).

Many thanks for the suggestion. We replaced Di to Dp,i throughout the text to avoid confusion.

P23, L15: "The results show a clear difference": I would suggest to slightly balance this assertion with

respect to Tir as, in a first approach, all groups (with the exception of G3 maybe) have relatively comparable median Tir, at least all belonging to the same range corresponding to mixed sky conditions.

*We agree with the reviewer's observation. We improved the text on the manuscript, which can be read as follows:*

*The results indicate differences between the four groups (G1 to G4) regarding the Tir and $\Delta\theta'e$ conditions during the event onset.*

P24, L7-8: When they speak of a stable Di, I assume that the authors mean that Di varies little from one event to another; if this is indeed the case, I would suggest not using the expression "remains stable", which, in my opinion, gives the impression of referring to a temporal evolution of the diameter. I would for example suggest "while the events belonging to G1 and G4 display comparable median Di".

*Thanks for the suggestion. Indeed, in the way it was written, the sentence could be confusing. The idea here was to show that both nocturnal groups, G1 and G4, present similar median Di on the event onset. We, thus, improved the sentence which now reads as follows:*

*The median GR for G1 and G4 varies from 3.8 to 4.2 nm h-1, respectively, and $CS_{growth}$ approximately constant around $1.0\times10^{-3}$ $s^{-1}$, while the median $D_{p,i}$ is similar for both nocturnal groups.*

P24, L14: Change "15 LT" to "15:00 LT" for consistency in the notation of times.

*Changed. Thanks.*

P24, L20: The value reported for G3 median Di in the text (24.3 nm) is slightly different from that reported in Table 2 (24.6 nm).

*Thanks for pointing out the typo. The correct value is 24.6 nm, which is now corrected in the manuscript.*

Fig. 3: Change "a particle growth event" to "2 particle growth events" in the figure caption.
Regarding the color bar, I would suggest indicating the concentrations in logarithmic scale instead of showing the logarithm of the concentrations, to facilitate the interpretation of the figure. If the authors wish to keep this representation, however, the unit should be corrected (log10(particle concentration) is not in cm-3). The same comment applies to similar figures.

*Many thanks for the suggestions. We implemented them and improved Fig. 3 and Fig.10 considering only dN/dlog10Dp. The caption of Fig. 3 now reads as:*

*Characteristic examples of two $CN_{<50}$ growth events at ATTO on 19 and 20 February 2018. The temporal evolution of the particle number size distribution (PNSD) is shown as a heat map, emphasizing the pronounced Aitken mode as well as particle growth events from the sub-50 nm particle to the Aitken mode during daylight.*

Fig. 7: Ticks on the x-axis are not located at the same place in all panels, which makes it difficult to read the times from the lowest panel.

*Thanks. We improved Figure 7 as suggested.*

Fig. S3: abbreviation "abr" for April should be changed to "apr".

Thanks. We replaced the figure with the correct abbreviation.

**References:**

[revised manuscript text omitted]

---

## Author Comment (AC2)

**Responses to the review of the manuscript "Occurrence and growth of sub-50 nm aerosol particles in the Amazonian boundary layer", by Franco et al., submitted for publication in ACP - manuscript ACPD 2021-765**

Dear editor, we would like to thank you and both reviewers for their valuable comments and useful suggestions to improve our manuscript. Below you can find answers and actions for each individual comment from the reviewers. In order to make it easier to identify the individual answers and actions, we used the following color code strategy:

- **In black, the referee's comments.**
- **In blue, the author's responses.**
- ***In blue and italic, the text modifications we made in the manuscript.***

**Responses to Anonymous Referee #2**

**Article "Occurrence and growth of sub-50 nm aerosol particles in the Amazonian boundary layer", by Franco et al., ACPD 2021-765**

Referee comment on "Occurrence and growth of sub-50 nm aerosol particles in the Amazonian boundary layer" by Marco A. Franco et al., Atmos. Chem. Phys. Discuss., https://doi.org/10.5194/acp-2021-765-RC2, 2021

This manuscript investigates the origin and growth of sub-50 nm particles over Amazonia. The presented analysis relies on several years of measurements in the boundary layer, extending earlier more campaign-based results on this topic. The paper is scientifically sound and, in most parts, very well written. I have a few, mostly minor issues to be considered further. After addressing these issues, the paper is, in my opinion, acceptable for publication in ACP.

We would like to thank referee #2 for the constructive comments and useful suggestions. They were helpful for clarifying important aspects of our analysis and discussion on the results.

The authors should provide some reasoning for selecting the periods G1 to G2 when looking at different times of the day. At first sight, it seems like this choice only covers 4 roughly equally long periods covering both nighttime (G1 and G4) and daytime (G2, G3). Furthermore, it seems evident that many of the differences observed between these 4 groups are simply the results of typical diurnal development of the boundary layer and its interactions with the rest of the troposphere. This feature should be explicitly brought up when discussing the results.

The choice of periods G1 to G4 was made due to particularities in the frequency of occurrence of growth events, as seen in Figure 7e. We have divided daytime events into two parts: those that occur in the morning, between 6:00 and 12:00, and those that occur in the afternoon. Morning events are those that are influenced by the evolution of the mixing layer, are mainly driven by photochemistry, and are the ones that occur in greater intensity. Afternoon events are mainly dominated by convective events. The nocturnal events were divided according to the evolution stage of the PBL. Events between 18:00 and 00:59 would be those that are still influenced by some kind of turbulence reminiscent of the mixing layer, but which may also still have had some influence from convective events. The growth events between 01:00 and 05:59 would be the most curious and they are not directly driven by photochemistry or variations in the PBL, since at this time the nocturnal PBL would already be well established. These events, in particular, occur under clear sky conditions and are the ones that raise the most questions about their origins.

In order to make the choice of periods explicit in the results sections, we have modified the text, which now reads as:

*To further investigate typical conditions or processes related to the observed growth events, we here discriminate between different groups based on their daily frequency distribution. The growth events (see Figure 7f) were divided into four groups (G1 to G4), where G1 and G4 represent nighttime and G2 and G3 represent daytime events. The daytime events were divided considering the occurrence of two frequency peaks: the first peak, representing 53 % of the growth events, is included in G2, covering the time from 06:00 to 11:59 LT. The afternoon increase, representing 21 % of the growth events, is included in G3, covering the time from 12:00 to 17:59 LT. The nocturnal events were divided according to the evolution stage of the PBL. Events between 18:00 and 00:59 (G4), with 10 % of the events are still influenced by some PBL turbulence, but may also have had some influence from convective events in the late afternoon. The growth events between 01:00 and 05:59 (G1), with 16 % of the total growth events are the most enigmatic ones. They are not directly driven by photochemistry or variations in the PBL, since at this time the nocturnal PBL is already well established. Different mechanisms such as air mass entrainment into the PBL by, e.g., intermittent turbulence (Dias-Júnior et al., 2017), nighttime rainfall events, or even an unknown biogenic source could play a role in the aerosol particle dynamics of in this time period.*

As a final note, was there some reason for having one hour (00:00-01:00) that was not included in any of the groups G1-G4?

The missing one hour was a typo, which we corrected in Table 2 of the revised manuscript. The correct label is 18:00 - 00:59.

Minor comments

While NPF may occur in both local and regional scales, occurrence of NPF leading to observable growth requires usually NPF taking place over relatively large spatial scales. Therefore, it somewhat misleading to claim that occurrence of NPF depends strongly on local conditions of individual sites (page 3, lines 3-4). Therefore, rather than emphasizing just local conditions, it should be noted that conditions (both emissions and meteorological conditions) over regional/synoptic scales are important in this respect.

Thanks for pointing that out. Indeed, NPF with the observable growth occurs at relatively large spatial scales and it should be more clear in the manuscript. Thus, we improved the text on P3, l 3-4 of the revised manuscript, which now reads as:

*The occurrence of NPF is dependent on the local conditions at individual sites, including meteorology, biogenic emissions, and air pollution levels, but regional and synoptic scales are also very important for this process. Particle growth events lasting on the order of hours are particularly influenced by larger geographic scales.*

Page 26, line 9: "early afternoon" does not match with the time period 06:00-11:00.

Thanks. We improved the text, which now reads as:

*During daytime, 53% of the growth events start in the morning between 06:00 and 11:00 00 LT, with a pronounced peak at 07:00 LT, showing a relation to the photochemistry and links with the evolution of the PBL.*

Page 26, lines 11-12: The author mention coagulation as an explanation for the observed decrease in N<50 and state that coagulation results in the growth of these particles. It is a relatively simple procedure to estimate the approximate growth rate of a mode of particles due to self-coagulation when knowing the particle number concentration in this mode. I recommend the authors to make this exercise. To me, it seems that for the typical values of N>50, coagulation can explain only a minor fraction of the observed growth of sub-50 nm particles, suggesting that this growth is mainly

due to condensation or other gas-to-particle conversion processes.

We agree with the reviewer, indeed the condensation plays a major role in the process, followed by a minor contribution of the coagulation process. Thus, we clarified the text, which now reads as follows:

*The subsequent decrease of $N_{<50}$ is likely due to condensation of semi- and less-volatile organic species on the sub-50 nm particles, resulting in growth.*

**References:**

Dias-Júnior, C. Q., Sá, L. D., Marques Filho, E. P., Santana, R. A., Mauder, M., and Manzi, A. O.: Turbulence regimes in the stable boundary layer above and within the Amazon forest, Agricultural and Forest Meteorology, 233, 122–132, https://doi.org/https://doi.org/10.1016/j.agrformet.2016.11.001, https://www.sciencedirect.com/science/article/pii/S0168192316304257, 2017.